# Interleukin 10 controls the balance between tolerance, pathogen elimination, and immunopathology in birds

Dominique Meunier[1†], Ricardo Corona-Torres[1†], Kay Boulton[1], Zhiguang Wu[1], Maeve Ballantyne[1], Laura Glendinning[1], Anum Ali Ahmad[1], Dominika Borowska[1], Lorna Taylor[1], Lonneke Vervelde[1‡], Jorge del Pozo[1], Marili Vasilogianni[2], José Jaramillo-Ortiz[2], Gonzalo Sanchez-Arsuaga[2], Androniki Psifidi[2], Fiona Tomley[2], Kellie A Watson[1], Michael J McGrew[1], Mark P Stevens[1], Damer P Blake[2], David A Hume[1*§]

[1]The Roslin Institute and Royal (Dick) School of Veterinary Studies, University of Edinburgh, Edinburgh, United Kingdom; [2]The Royal Veterinary College, London, United Kingdom

*For correspondence:
david.hume@uq.edu.au

†These authors contributed equally to this work

Present address: ‡Royal GD Animal Health, Deventer, Netherlands; §Mater Research Institute-University of Queensland, Brisbane, Australia

## eLife Assessment

IL10 balances protective and deleterious immune functions in mice and humans, but if IL10 also controls avian intestinal homeostasis remains unclear. Generating genetic knockouts, Meunier et al. established that a complete lack of IL10 strengthened immunity against enteric bacteria in chickens, while also aggravating infection-inflicted inflammatory tissue damage and dysbiosis upon parasite infection, but unlike mouse models, IL10 deficiency did not provoke spontaneous colitis in chickens. The findings presented are **valuable**, and the strength of evidence is **convincing**. The observation may have implications for the livestock industry and additional studies involving genetic knockouts may further unravel conserved and distinct avian IL10 controls.

**Abstract** Effective mucosal immunity in the intestine involves a fine balance between tolerance of the microbiome, recognition, and elimination of pathogens, and inflammatory tissue injury. The anti-inflammatory cytokine IL10 regulates these processes in the intestines of mice and humans; the anti-inflammatory activity of IL10 is also conserved in birds. To determine the function of IL10 in avian mucosal immunity, we generated germ line modifications of the chicken *IL10* locus to abolish or reduce IL10 expression. *In vitro* analysis of macrophage response to lipopolysaccharide confirmed the loss of IL10 protein expression, the lack of dosage compensation in heterozygotes, and prevention of autocrine inhibition of nitric oxide production in homozygous IL10 knockout macrophages. IL10-deficiency significantly altered the composition of the caecal microbiome, but unlike IL10-deficient mice and humans, IL10-deficient chickens did not exhibit spontaneous colitis. Following experimental challenge with *Salmonella enterica* serovar Typhimurium or *Campylobacter jejuni* in IL10-deficient chickens, enhanced clearance of the pathogens was associated with elevated transcription of pro-inflammatory genes and increased infiltration of inflammatory cells into gut mucosa. In IL10-deficient chickens challenged with the parasite *Eimeria tenella*, pathogen clearance was accelerated but caecal lesions were more severe and weight gain was compromised. Neither the heterozygous IL10 knockout nor a homozygous IL10 enhancer mutation had a major effect on pathogen clearance or inflammation in any of the challenge models. Our findings highlight the intrinsic compromise in mucosal immune response and have important implications for the development of strategies to combat avian and zoonotic pathogens in poultry.

## Introduction

Elimination of potential pathogens by the innate and acquired immune systems involves an intrinsic compromise between host protection and tissue injury that requires mechanisms to control the nature, magnitude, duration, and specificity of effector pathways. A key component of specificity is the capacity to distinguish between pathogens and the commensal microbiome, and nowhere is this challenge more pressing than in the intestine. The anti-inflammatory cytokine, interleukin 10 (IL10), has long been recognised as an essential feedback regulator of intestinal inflammation in mammals (reviewed in *Saraiva et al., 2020*). IL10-deficient mice generated by homologous recombination developed spontaneous enterocolitis (*Kühn et al., 1993*). Subsequent studies revealed that components of the intestinal microbiome were essential to the development of pathology and determined the extent and location of lesions (*Sellon et al., 1998*). In humans, homozygous mutations in genes encoding IL10 or its receptor, IL10RA, have been associated with early-onset inflammatory bowel disease (*Saraiva et al., 2020*).

In mammals, IL10 is produced by both myeloid and T and B lymphoid cells following activation by immune stimuli (*Saraiva et al., 2020*; *Couper et al., 2008*), with subtle differences between mice and humans (*Rasquinha et al., 2021*). In the specific context of the intestine, conditional deletion of the *IL10* and *IL10RA* genes in mice indicated that production of the ligand by macrophages was not required for pathology, consistent with T cells being the major source of intestinal IL10 (*Brockmann et al., 2018*; *Pils et al., 2011*). By contrast, macrophage-specific deletion of the *IL10RA* gene encoding the IL10 receptor was sufficient to elicit spontaneous colitis in mice (*Bernshtein et al., 2019*; *Redhu et al., 2017*; *Zigmond et al., 2014*). The contribution of IL10 to feedback modulation of host defence is not restricted to the gut. There have been countless studies of the impact of IL10 deficiency in mouse parasite, bacterial, fungal, and viral disease models (reviewed in *Couper et al., 2008*). The primary focus of these studies has been on the mitigation of immunopathology by IL10, with less attention given to pathogen elimination. By contrast to studies in mice, analysis of the quantitative importance of IL10 in other species has been largely correlative. Both ligand and receptor are conserved in vertebrates, and zebrafish has been discussed as a model for understanding IL10 biology (*Piazzon et al., 2016*). Here, we focus on the chicken, both a model vertebrate and an economically important livestock species.

The original cloning and sequencing of chicken *IL10* cDNA and genomic DNA (*Rothwell et al., 2004*) described the immunomodulatory activity of the recombinant protein on T cell production of interferon-gamma (IFNγ), its regulated expression in T cells and macrophages *in vitro*, and increased mRNA expression during parasite (*Eimeria maxima*) infection. Like the mammalian transcript, chicken *IL10* mRNA contains multiple copies of the AUUUA instability motif in the 3'-untranslated region. We and others have provided evidence that quantitative variation in IL10 production may contribute to inter-individual or inter-breed variation in infectious disease susceptibility in chickens (*Russell et al., 2021*; *Boulton et al., 2018a*; *Swaggerty et al., 2015*; *Calenge et al., 2009*; *Ghebremicael et al., 2008*). To test this hypothesis directly, and more generally to determine the conservation of IL10 biological function across vertebrates, we have generated germ line modifications of the chicken *IL10* locus that abolish or reduce IL10 expression. We describe the lack of dosage compensation in heterozygous mutation and present evidence that the reduced expression of IL10 leads to increased pathogen clearance at the expense of increased inflammation.

## Results

### Generation of IL10-deficient chickens

The emergence of CRISPR/Cas9 technology and the ability to culture, cryopreserve, genetically modify, and subsequently transfer chicken primordial germ cells (PGCs) into sterile surrogate host embryos has expedited the analysis of gene function in the chick (*Figure 1A*; *Ballantyne et al., 2021a*; *Ioannidis et al., 2021*; *Wu et al., 2023*; *Idoko-Akoh and McGrew, 2023*). We used this approach to edit the *IL10* locus in chicken PGCs and create two independent lines of IL10-deficient chickens (*Figure 1*). We first created an IL10 knockout (IL10KO) line by introducing a premature in-frame stop codon in exon 1 of the *IL10* gene in chicken PGCs, 75 bp downstream of the start codon (C26>*; *Figure 1B and D*, *Figure 1—figure supplement 1*). To generate a potential expression hypomorph, we created an IL10-Enhancer knockout (IL10EnKO) line by deleting a 533 bp genomic fragment encompassing a

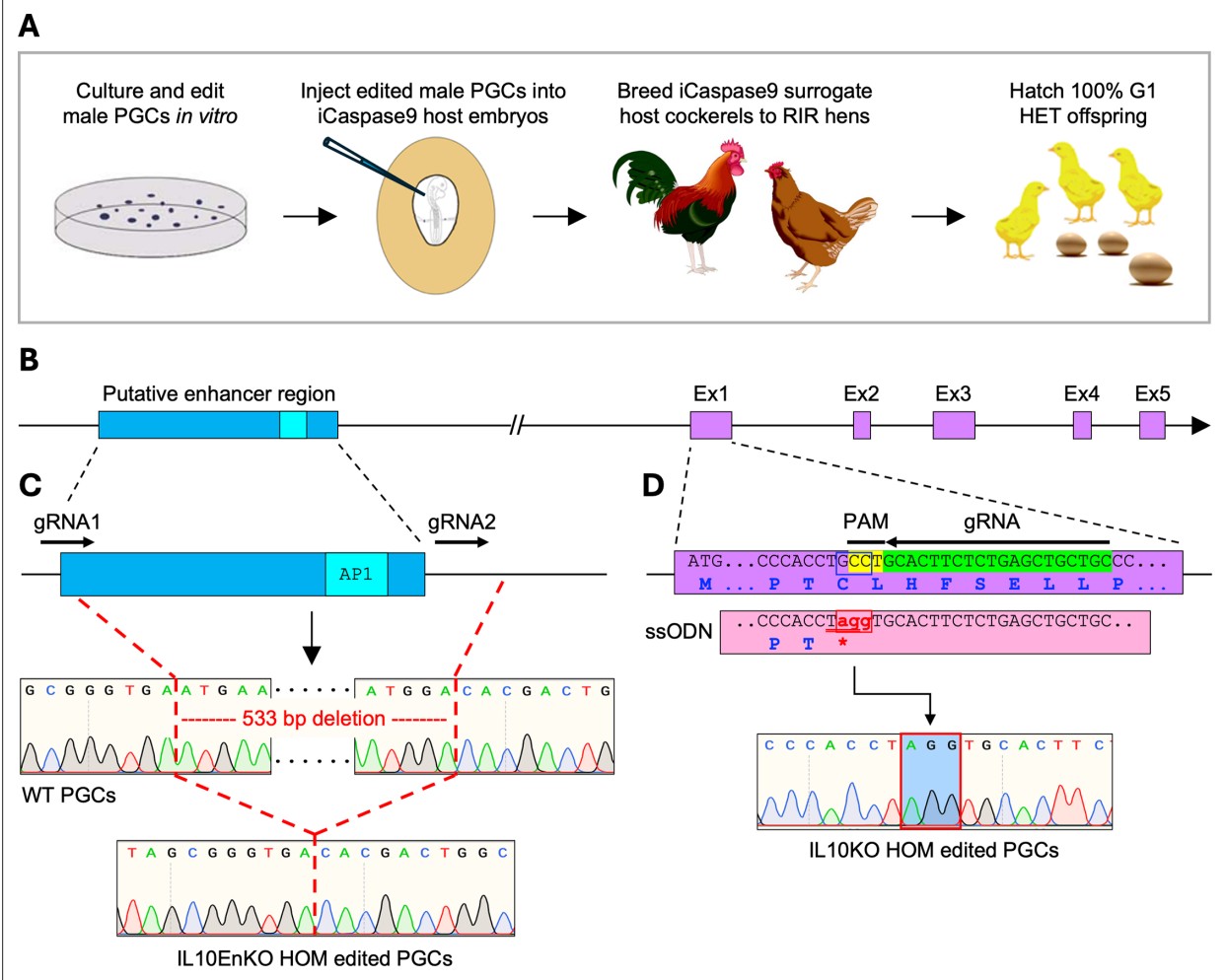

**Figure 1.** IL10 knockout (IL10KO) and IL10-Enhancer knockout (IL10EnKO) editing strategies. (**A**) Schematic of the pipeline used to create gene-edited chickens using embryonic primordial germ cells (PGCs) and the iCaspase9 sterile surrogate host line. Male PGCs carrying the desired biallelic (homozygous [HOM]) edits were injected into iCaspase9 sterile surrogate host embryos to create IL10KO and IL10EnKO chickens. Upon sexual maturity, surrogate host cockerels were bred to RIR hens, resulting in 100% IL10KO heterozygous (HET) or IL10EnKO HET offspring in the first generation (G1). (**B**) Schematic of the *IL10* locus depicting the *IL10* putative enhancer region and the five *IL10* exons (Ex1 to Ex5). (**C**) IL10EnKO HOM PGCs were created by deleting a 533 bp fragment encompassing the *IL10* putative enhancer region, using two guide RNAs (gRNAs); DNA sequencing confirmed the biallelic deletion. (**D**) IL10KO HOM PGCs were created using one gRNA and a 143 bp repair template (ssODN) to modify three nucleotides in *IL10* exon 1 (red, lowercase), thus introducing a premature in-frame stop codon in the *IL10* gene (underlined, *); DNA sequencing confirmed the biallelic edit.

The online version of this article includes the following source data and figure supplement(s) for figure 1:

**Figure supplement 1.** Wild-type and edited IL10 sequences for IL10 knockout (IL10KO) edit.

**Figure supplement 2.** Wild-type and edited *IL10* putative enhancer sequences for IL10-Enhancer knockout (IL10EnKO) edit.

**Figure supplement 3.** Alignment of conserved enhancer-like sequences in the *IL10* locus of different avian species.

**Figure supplement 4.** Genotyping strategy for IL10 knockout (IL10KO) and IL10-Enhancer knockout (IL10EnKO) edited primordial germ cells (PGCs) and birds.

**Figure supplement 4—source data 1.** PDF file containing the annotated raw, unedited, and uncropped agarose gel images shown in *Figure 1—figure supplement 4*, with the relevant bands clearly labelled.

**Figure supplement 4—source data 2.** Original files of the raw, unedited, and uncropped agarose gel images shown in *Figure 1—figure supplement 4*.

non-coding sequence located approximately 2.3–2.8 kb upstream of the *IL10* transcription start site, in chicken PGCs (*Figure 1B and C*, *Figure 1—figure supplement 2*). This sequence was targeted based upon alignment of sequences from multiple avian species that identified a core of 300 bp with conservation around 85%, containing perfectly conserved motifs for known transcriptional regulators,

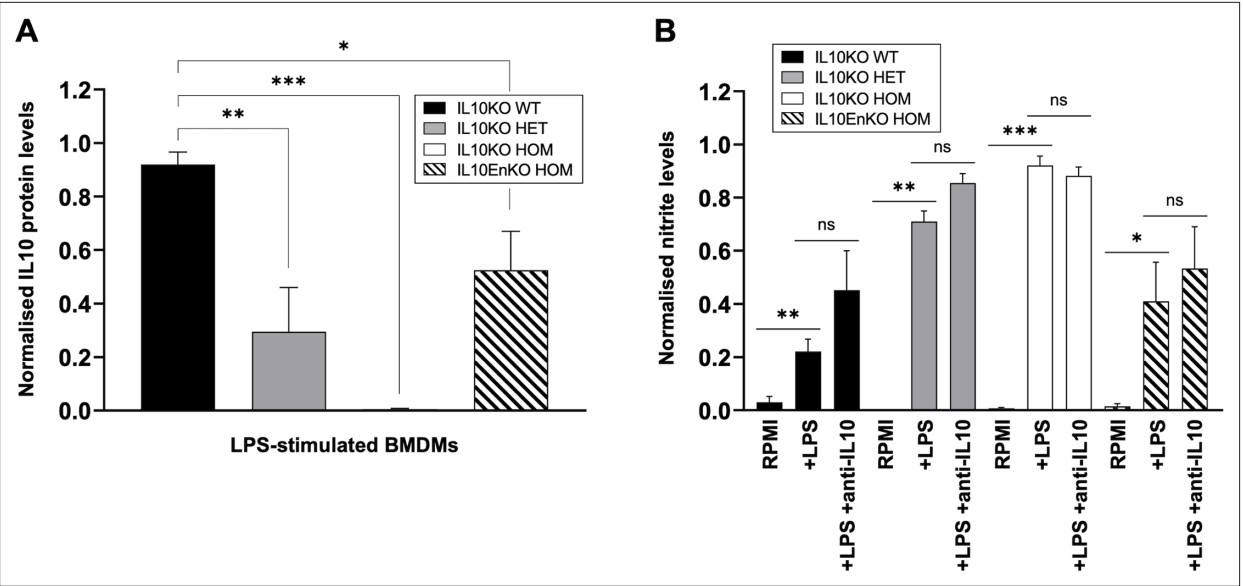

**Figure 2.** Confirmation of the IL10 mutation in bone marrow-derived macrophages (BMDMs). (**A**) IL10 protein levels were measured by capture ELISA for lipopolysaccharide (LPS)-stimulated BMDMs derived from day 18 IL10 knockout (IL10KO) wild-type (WT) (n=4), HET (n=2), homozygous (HOM) (n=4), and IL10-Enhancer knockout (IL10EnKO) HOM (n=4) embryos. Two independent experiments were performed, and protein levels were normalised against the highest IL10KO WT value for each experiment and then combined. LPS treatment induced IL10 production in IL10KO WT BMDMs, but not in IL10KO HOM BMDMs; intermediate levels of IL10 were detected in IL10KO HET and IL10EnKO HOM BMDMs; IL10 expression in non-LPS-induced samples was negligible (<0.015; not shown). (**B**) Nitric oxide production was assessed by measuring nitrite levels using Griess assay for LPS-stimulated BMDMs derived from day 18 IL10KO WT (n=6), HET (n=2), HOM (n=7), and IL10EnKO HOM (n=6) embryos, in the absence or presence of neutralising anti-IL10 antibody ROS-AV163 (anti-IL10). Three independent experiments were performed and nitrite levels were normalised against the highest IL10KO HOM nitrite value for each experiment before being combined. A significant increase in nitrite levels was observed in the absence of IL10 in LPS-stimulated IL10KO HOM BMDMs; nitrite levels were also significantly increased in LPS-stimulated IL10KO HET and IL10EnKO HOM BMDMs. Addition of neutralising anti-IL10 antibody did not significantly affect nitrite levels in LPS-stimulated BMDMs, independently of the IL10 genotype. Note that each BMDM sample was derived from three pooled embryos of the same genotype. Data displayed as mean with SE. Statistical significance calculated using two-tailed unpaired t tests; *p<0.05, **p<0.01, ***p<0.001, ns: not significant.

including AP1, FOXP3, and members of the NFAT/ETS family (*Figure 1—figure supplement 3*). The size and species conservation of this element resembles the well-characterised intronic enhancer of the *Csf1r* locus (*Hume et al., 2017*).

To create IL10KO and IL10EnKO chickens, male PGCs carrying the desired biallelic edit in *IL10* exon 1 or in the putative *IL10* enhancer region (as validated by PCR and DNA sequencing; *Figure 1C and D*, *Figure 1—figure supplement 4*) were injected into iCaspase9 sterile surrogate host embryos, in which a chemically inducible caspase 9 protein is expressed from the PGC-specific *DAZL* locus, together with a GFP reporter (*Ballantyne et al., 2021b*). Embryos were treated with the iCaspase9 activation chemical AP20187, incubated to hatch, and the resulting chicks were raised to sexual maturity under specified pathogen-free (SPF) conditions at the National Avian Research Facility (NARF). Mating of the iCaspase9 surrogate host cockerels to Rhode Island Red (RIR) hens generated 100% IL10KO or IL10EnKO heterozygous (HET) offspring in the first generation (G1; *Figure 1A*; see *Supplementary file 1* for detailed G1 bird numbers). Upon sexual maturity, G1 HET birds were intercrossed to generate the second generation (G2) of IL10KO and IL10EnKO wild-type (WT), HET, and homozygous (HOM) birds for analyses. Both lines produced WT, HET, and HOM offspring at a 1:2:1 Mendelian ratio (see *Supplementary file 1* for detailed G2 bird numbers).

## Confirmation of the IL10 mutation in bone marrow-derived macrophages

Chicken bone marrow-derived macrophages (BMDMs) cultured in recombinant macrophage colony-stimulating factor respond to bacterial lipopolysaccharide (LPS) by secreting IL10 that can be detected by ELISA (*Wu et al., 2016*). The autocrine function of IL10 in this assay was demonstrated by the addition of neutralising anti-IL10 antibody, which amplified the expression of nitric oxide synthase

(NOS2) and production of nitric oxide (*Wu et al., 2016*). *Figure 2A* confirms the release of IL10 by LPS-stimulated BMDMs derived from WT embryos and the complete absence of detectable IL10 release by BMDMs derived from IL10KO HOM embryos. The absence of IL10 was associated with a significant increase in nitric oxide production that was not additive with the impact of added anti-IL10 antibody (*Figure 2B*). Since IL10 mediates feedback control, it is conceivable that the loss of one functional allele could be compensated by increased expression from the other allele. However, we found that expression of IL10 was reduced by up to 50% in BMDMs derived from IL10KO HET embryos, with a corresponding partial impact on nitric oxide production (*Figure 2A and B*). The impact of the IL10EnKO HOM mutation was more variable between samples in both assays, but consistent with a partial expression hypomorph (*Figure 2A and B*).

## Loss of IL10 does not lead to spontaneous immunopathology

The IL10KO and IL10EnKO chicken lines were created and raised under SPF conditions in the NARF SPF avian facility. Close monitoring of the WT, HET, and HOM birds over several months in that environment did not reveal any health issues or adverse phenotypes. The IL10KO and IL10EnKO edits did not significantly affect the post-hatch growth of HET and HOM birds compared to their respective WT controls (*Figure 3A and B*), although IL10EnKO HET and HOM hens were overall heavier than IL10EnKO WT controls (*Figure 3B*). Regular gross post-mortem examination of tissues and histopathological analyses of the gastrointestinal tract over several months, using a semiquantitative scoring system (*McCafferty et al., 2000*), revealed no evidence of intestinal pathology in IL10KO HOM and IL10EnKO HOM birds compared to WT controls (*Figure 3C*). These results are in contrast with the early onset colitis and failure to thrive phenotypes observed in IL10-deficient mouse models in both conventional and SPF environments (*Kühn et al., 1993*; *Redhu et al., 2017*; *Papoutsopoulou et al., 2021*; *Büchler et al., 2012*).

Spontaneous colitis in *Il10*$^{-/-}$ mice is associated with and dependent upon dysbiosis in the intestinal microbiome (*Devkota et al., 2012*; *Gunasekera et al., 2020*; *Keubler et al., 2015*). We compared the caecal microbiome of IL10KO WT and HOM birds at 4 weeks of age using 16S rDNA V3-V4 hypervariable amplicon sequencing as described (*Pandit et al., 2018*; *Salavati Schmitz et al., 2024*). The caecal microbiome of both genotypes was dominated by the phylum Firmicutes (*Figure 4A*) and the most abundant genera were *Lactobacillus* and *Faecalibacterium* (*Figure 4B*). Linear discriminant analysis (LDA) effect size (LEfSe) analysis showed enrichment of the phylum Firmicutes and the genera *Defluviitaleaceae_UCG-011* and *Paludicola* in the IL10KO HOM group, while the phylum Bacteroidota and the genera *Alistipes* and *UCG_008* were enriched in the WT group (*Figure 4C*). Principal component analyses showed clear separation of the microbial communities based on beta diversity (*Figure 4D*; $R^2$=0.25, p=0.017). These findings confirm that mucosal IL10 regulates the composition of the chicken microbiome.

To evaluate a potential environmental impact on the phenotype of IL10-deficient birds, a cohort of IL10KO WT and HOM birds was hatched and raised in the NARF conventional avian facility. IL10KO HOM birds remained healthy in that environment as they aged. Post-growth hatch rates were comparable between IL10KO WT and HOM hens, with the exception of a marginal effect on growth that resolved with age (*Figure 3D*); there was also no evidence of overt intestinal pathology in IL10KO HOM birds compared to WT controls (*Figure 3E*). Cellular infiltration scores were, however, significantly higher in the duodenum and proximal colon of both IL10KO WT and HOM birds raised in the conventional facility as opposed to the SPF environment (*Figure 3F*).

Two important differences exist between the IL10KO mouse and chicken models. Firstly, the birds are outbred. Most mouse studies are conducted on an inbred background, and the impact of the *IL10* mutation is strain-specific (*Büchler et al., 2012*). Secondly, chickens in the NARF conventional avian facility are routinely immunised against major avian pathogens (*Supplementary file 2*). The original analysis of IL10-deficient mice revealed little impact on T cell-dependent antibody responses (*Kühn et al., 1993*). We measured antibody titres in 29-week-old IL10KO WT and HOM birds raised in the NARF conventional facility for seven of the pathogens that the birds were vaccinated against (*Figure 5*): avian encephalomyelitis virus (AEV), chicken anaemia virus (CAV), duck adenovirus (the agent of egg drop syndrome [EDS]), infectious bursal disease virus (IBDV), infectious bronchitis virus (IBV), infectious laryngotracheitis virus (ILTV), and Newcastle disease virus (NDV). The administration schedule, nature, and delivery mode of these vaccines is summarised in *Supplementary file 2*.

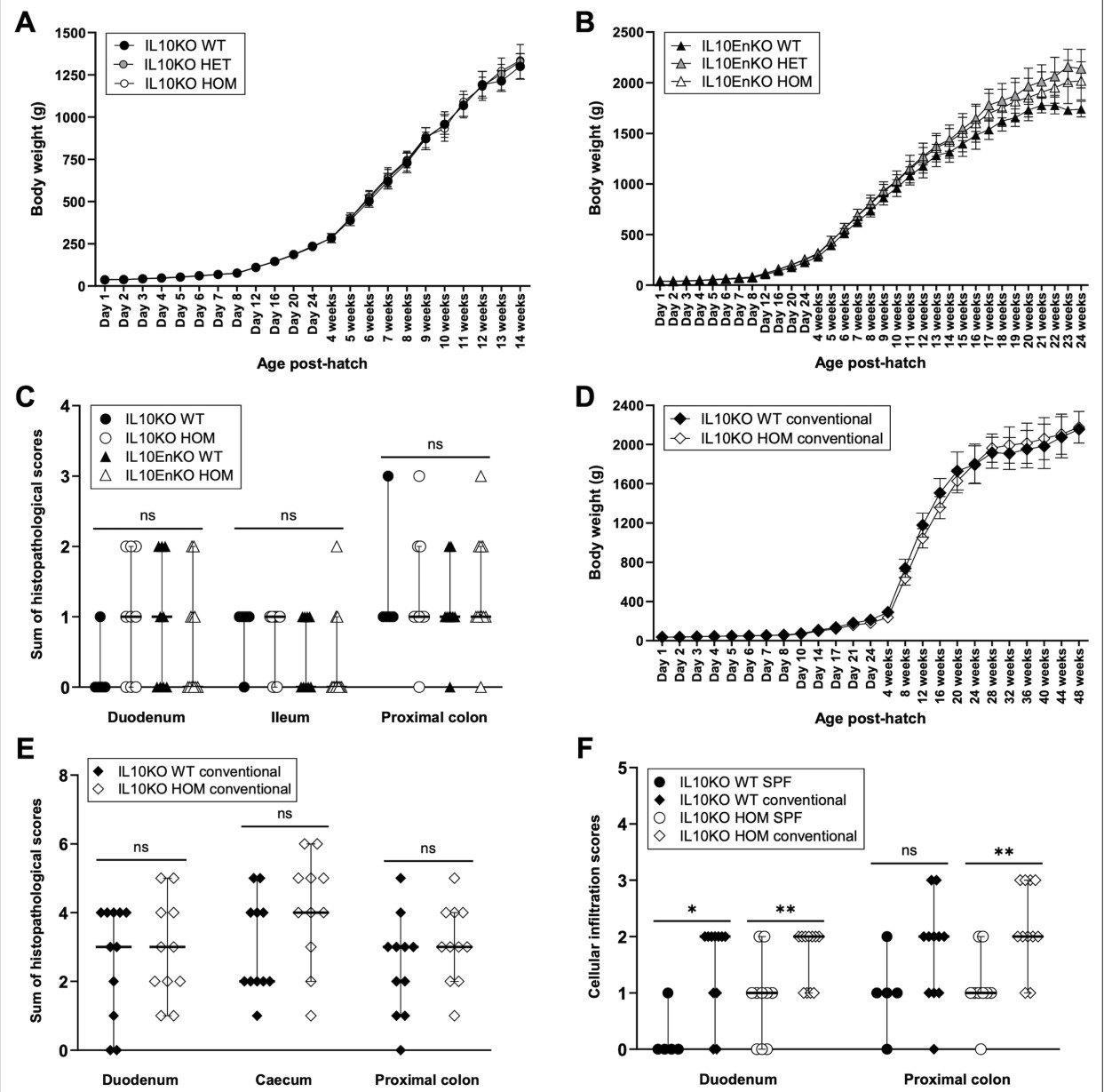

**Figure 3.** Growth curves and histopathological analyses of the gastrointestinal tract of IL10-deficient chickens in specified pathogen-free (SPF) and conventional facilities. (**A**) Growth curves for IL10 knockout (IL10KO) wild-type (WT) (n=3–8), heterozygous (HET) (n=11–16), and homozygous (HOM) (n=2–4) hens raised in the National Avian Research Facility (NARF) SPF facility, from hatch to 14 weeks. No significant weight differences were observed between genotypes at any timepoint (p>0.05). (**B**) Growth curves for IL10-Enhancer knockout (IL10EnKO) WT (n=2–4), HET (n=7–11), and HOM (n=5–11) hens raised in the NARF SPF facility, from hatch to 24 weeks. IL10EnKO HET and HOM hens were overall heavier than WT controls, but this was statistically significant only between day 8 and day 24 (p<0.05). (**C**) Sums of histopathological scores for IL10KO WT (n=5) and HOM (n=9), and for IL10EnKO WT (n=9) and HOM (n=11) birds raised in the NARF SPF facility; tissue samples were collected at regular intervals from 16 to 40 weeks post-hatch. No significant differences were observed between genotypes for the tissues analysed. (**D**) Growth curves for IL10KO WT (n=2–10) and HOM (n=5–15) hens raised in the NARF conventional facility, from hatch to 48 weeks. IL10KO HOM hens were significantly smaller than WT controls from 3 to 19 weeks post-hatch (p<0.05), but this difference resolved with age. (**E**) Sums of histopathological scores for IL10KO WT and HOM birds (n=11 in each group) raised in the NARF conventional facility; tissue samples were collected at regular intervals from 10 to 50 weeks post-hatch. No significant differences were observed between genotypes for the tissues analysed. (**F**) Cellular infiltration scores for IL10KO WT and HOM birds raised in the NARF SPF and conventional facilities (same bird numbers as in C and E). Cellular infiltration scores were overall significantly higher for IL10KO WT and HOM birds raised in the conventional facility. Maximum possible sums of scores for data in C and E=11; score range for data in F=0–3. Data displayed as mean with SD (**A, B, D**) or as median with 95% confidence interval (**C, E, F**). Statistical significance calculated using one-way ANOVA with Bonferroni multiple comparison tests (**A, B**), Kruskal-Wallis test (**C**), two-tailed unpaired t tests (**D**), or Mann-Whitney U tests (**E, F**); *p<0.05, **p<0.01, ns: not significant.

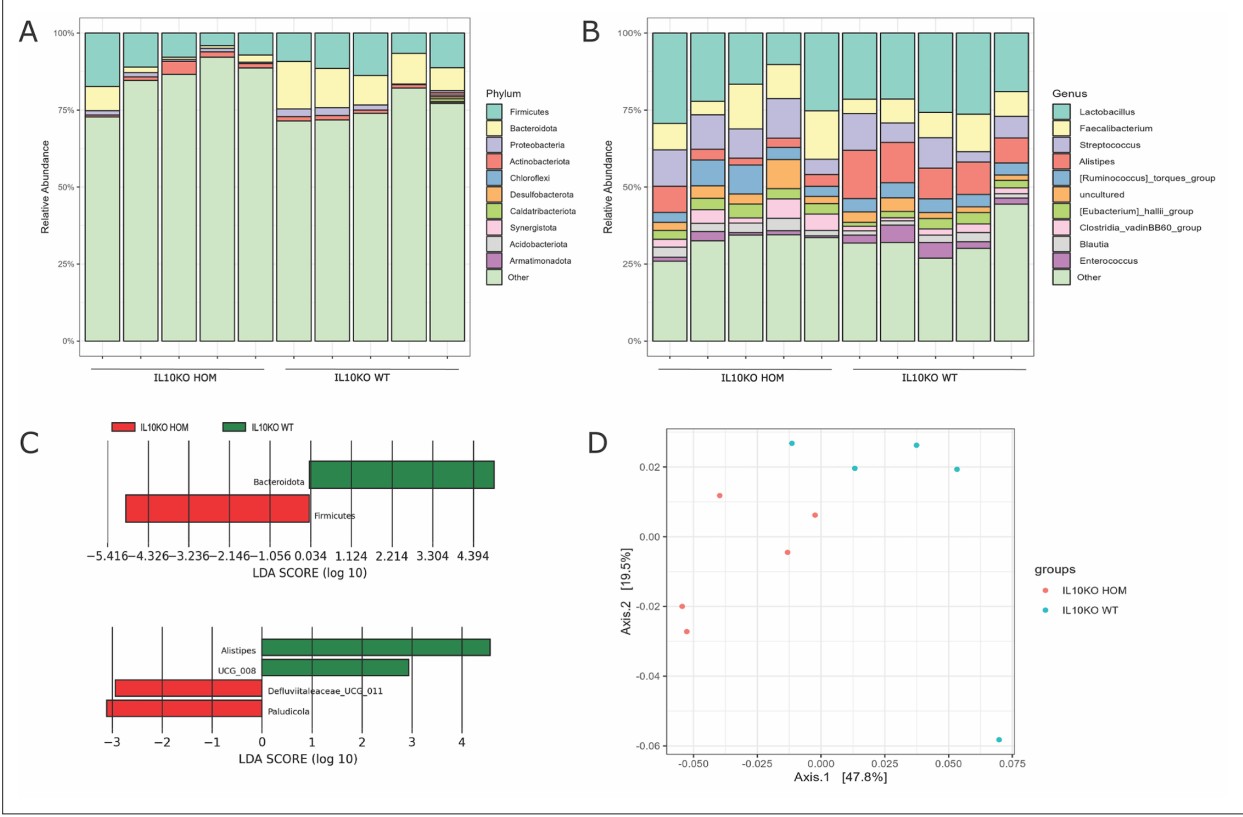

**Figure 4.** Impact of IL10 knockout (IL10KO) homozygous (HOM) mutation on the caecal microbiota of chickens. (**A–B**) Sequencing of 16S rDNA variable regions amplified from DNA extracted from the caecal contents of IL10KO wild-type (WT) and HOM birds (n=5 in each group) at 4 weeks of age revealed differences in the relative abundance of microbial phyla (**A**) and genera (**B**). Each vertical bar shows data from a different bird. (**C**) Linear discriminant analysis (LDA) effect size (LefSe) analysis identified differentially abundant taxa between IL10KO WT and HOM birds. (**D**) Principal component analyses based on weighted Unifrac distance showed clear clustering of samples (Adonis2 $R^2$=0.25, p=0.017). Axis 1: first principal component, Axis 2: second principal component; value in brackets: total percentage of variation between samples.

Statistical analyses revealed differences in vaccine response between IL10KO WT and HOM birds. Antibody titres to AEV and IBDV were significantly lower in IL10KO HOM birds compared to WT controls, whereas those to EDS and IBV were significantly higher in IL10KO HOM birds compared to WT controls (*Figure 5*). AEV and IBDV vaccination were carried out with live vaccines delivered in the drinking water, whereas EDS and the last dose of IBV vaccination were done with inactivated vaccines given by intramuscular injection (*Supplementary file 2*). Antibody titres to CAV, ILTV, and NDV were not significantly different between IL10KO HOM birds and WT controls (*Figure 5*).

## The impact of IL10 deficiency on the response to enteric bacterial infection

*Campylobacter jejuni* (*C. jejuni*) and non-typhoidal *Salmonella* are zoonotic diarrhoeal pathogens of global importance (*Havelaar, 2010*). Breed-specific pathology associated with *C. jejuni* infection in chickens was associated with relatively low caecal *IL10* mRNA (*Humphrey et al., 2014*). Similarly, IL10-deficient C57BL/6J mice exhibited severe inflammation in the caecum and colon upon *C. jejuni* infection, whereas congenic WT mice were persistently colonised in the absence of gut pathology (*Mansfield et al., 2007*). A similar outcome was reported in IL10-deficient mice infected with *Salmonella* Typhimurium (*S.* Typhimurium) (*Schultz et al., 2018*).

To study the role of IL10 during *Campylobacter* infection in chickens, two cohorts of 20 IL10KO WT, HET, or HOM chickens obtained from the NARF SPF avian facility were inoculated at 14 days of age with $10^2$ colony forming units (CFU) *C. jejuni* strain M1 and analysed 1 and 2 weeks post-infection. The dose of strain M1 used is the lowest needed to achieve reliable caecal colonisation (*Vohra et al., 2020*). In the first cohort, colonisation of the caeca by *C. jejuni* in IL10KO HOM birds was significantly

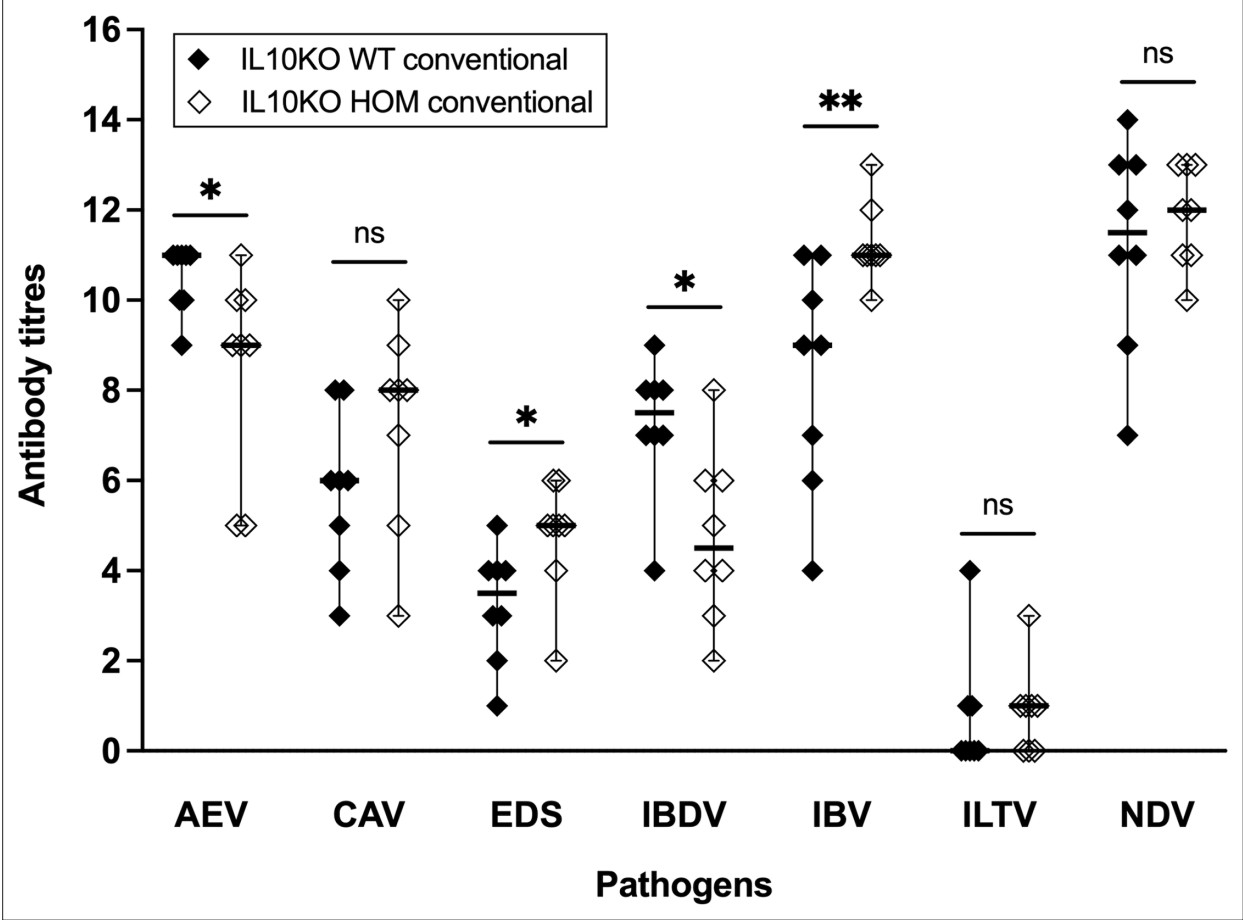

**Figure 5.** Response to vaccination in IL10 knockout (IL10KO) wild-type (WT) and homozygous (HOM) chickens. Blood samples were collected from 29-week-old IL10KO WT and HOM vaccinated hens raised in the National Avian Research Facility (NARF) conventional facility (n=8 in each group), and antibody titres were measured by ELISA. Titres to avian encephalomyelitis virus (AEV) and infectious bursal disease virus (IBDV) were significantly lower in IL10KO HOM hens compared to WT controls, whereas titres to duck adenovirus (the agent of egg drop syndrome [EDS]) and infectious bronchitis virus (IBV) were significantly higher in IL10KO HOM hens compared to WT controls; titres to chicken anaemia virus (CAV), infectious laryngotracheitis virus (ILTV), and Newcastle disease virus (NDV) were not significantly different between IL10KO HOM and WT hens. Data displayed as median with 95% confidence interval. Statistical significance calculated using Mann-Whitney U tests; *p<0.05, **p<0.01, ns: not significant.

lower at 1 week post-infection, with a median of 4.8 $\log_{10}$ CFU $g^{-1}$ compared to >9 $\log_{10}$ CFU $g^{-1}$ in WT birds (*Figure 6A*). However, this was not sustained and by 2 weeks post-infection, all groups had caecal colonisation at >9 $\log_{10}$ CFU $g^{-1}$ (*Figure 6A*). In the second cohort, median counts were not statistically different between IL10KO WT and HOM birds at 1 week post-infection, but caecal colonisation decreased by ~2 $\log_{10}$ CFU $g^{-1}$ in IL10KO HOM birds at 2 weeks post-infection relative to WT birds (*Figure 6A*). The IL10KO HET birds did not differ significantly from WT in either cohort (*Figure 6A*). Histopathological analysis revealed that the caecal mucosa of IL10KO HOM birds presents with significantly increased numbers of lymphocytes and plasma cells, admixed with infiltrating heterophils, when compared to WT and IL10KO HET birds in both cohorts (*Figure 6—figure supplements 1 and 2*). To confirm the apparent hyperinflammatory response in IL10KO HOM birds, we used multiplex quantitative reverse transcriptase-PCR (qRT-PCR) to measure expression of immunity-related genes in the caecum, as described (*Borowska et al., 2019*). In both cohorts, we detected large increases in expression of pro-inflammatory cytokines, chemokines, and other effectors, notably *Nos2*, in the IL10KO HOM caecum relative to WT (*Figure 6B*).

To test the role of IL10 in response to *Salmonella* infection, groups of 20 IL10KO WT, HET, or HOM chickens were inoculated at 14 days of age with $10^8$ CFU *S.* Typhimurium strain ST4/74 nal[R] (*Chaudhuri et al., 2013*). The impact of lack of IL10 on bacterial burden in caecum, liver, and spleen was marginal at 1 week, but by 2 weeks the majority of IL10KO HOM birds had eliminated the pathogen,

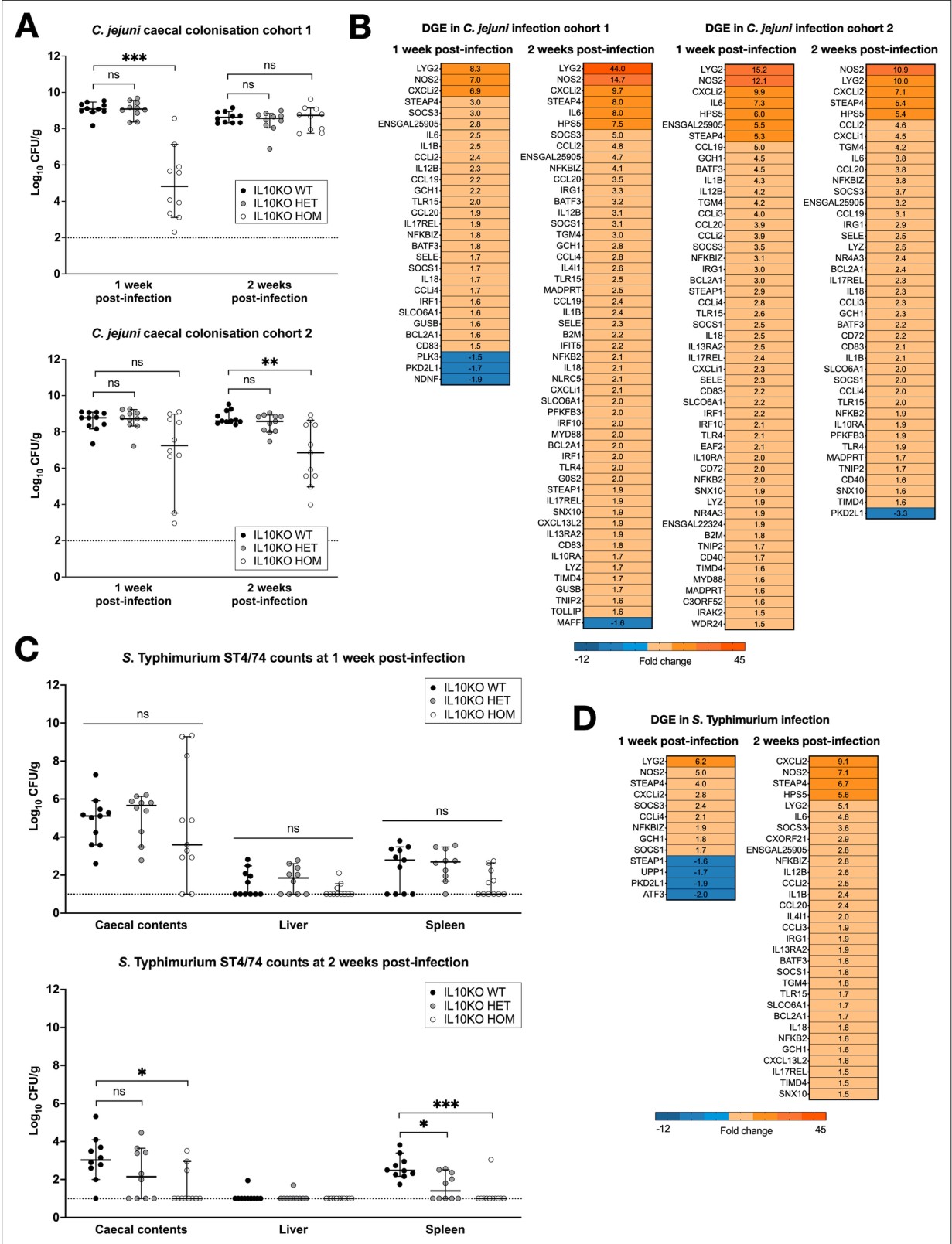

**Figure 6.** Net replication of *C. jejuni* and *S.* Typhimurium in the caeca of IL10 knockout (IL10KO) wild-type (WT), heterozygous (HET), and homozygous (HOM) chickens and immune responses to infection. (**A**) Caecal burden of *C. jejuni* strain M1 in IL1KO WT, HET, and HOM chickens at 1 and 2 weeks post-inoculation in two separate trials. (**B**) Heat map of differentially transcribed genes measured by multiplex quantitative reverse transcriptase-PCR (qRT-PCR) in the caeca of IL10KO HOM chickens in relation to IL10KO WT birds at 1 and 2 weeks post-inoculation in two separate trials. An increase in

*Figure 6 continued on next page*

*Figure 6 continued*

the expression of pro-inflammatory genes was consistently observed in both studies. (**C**) Burden of *S.* Typhimurium strain ST4/74 in the caeca, liver, and spleen of IL10KO WT, HET, and HOM chickens at 1 and 2 weeks post-inoculation (single trial). (**D**) Heat map of differentially transcribed genes in the caeca of IL10KO WT and HOM chickens 1 and 2 weeks post-inoculation with ST4/74. All group sizes: n=10 or 11. Bacterial colonisation shown as median with 95% confidence interval; dotted line shows limit of detection for each study (**A, C**). Statistical significance calculated using one-way ANOVA with Kruskal-Wallis test followed by Dunn's multiple comparison tests; *p<0.05, **p<0.01, ***p<0.001, ns: not significant.

The online version of this article includes the following figure supplement(s) for figure 6:

**Figure supplement 1.** Histopathological findings in the ileum and caecum of IL10 knockout (IL10KO) wild-type (WT), heterozygous (HET), and homozygous (HOM) chickens infected with *C. jejuni* cohort 1.

**Figure supplement 2.** Histopathological findings in the ileum and caecum of IL10 knockout (IL10KO) wild-type (WT), heterozygous (HET), and homozygous (HOM) chickens infected with *C. jejuni* cohort 2.

**Figure supplement 3.** Histopathological findings in the ileum and caecum of IL10 knockout (IL10KO) wild-type (WT), heterozygous (HET), and homozygous (HOM) chickens infected with *S.* Typhimurium.

**Figure supplement 4.** Differential gene expression in the caecum of IL10 knockout (IL10KO) wild-type (WT) and homozygous (HOM) chickens infected with *C. jejuni*.

**Figure supplement 5.** Differential gene expression in the caecum of IL10 knockout (IL10KO) wild-type (WT) and homozygous (HOM) chickens infected with *S.* Typhimurium at 1 week (**A**) and 2 weeks (**B**) post-infection.

whereas WT birds retained bacteria in both caecum and spleen (*Figure 6C*). Enhanced lymphoplasmacytic and heterophil infiltration was observed in the ileum and caecum of IL10KO HOM compared to WT or IL10KO HET birds at both intervals after *S.* Typhimurium infection (*Figure 6—figure supplement 3*). As observed following the *C. jejuni* challenge, analysis of inflammatory gene expression by multiplex qRT-PCR revealed a substantial increase in response in the IL10KO HOM birds relative to WT (*Figure 6D*). Volcano plots showing the direction and magnitude of gene expression differences detected between IL10KO WT and HOM birds following bacterial infection are shown in *Figure 6—figure supplements 4 and 5*. In summary, the absence of IL10 leads to a sustained increase in inflammation in response to enteric bacterial infection that is also associated with more effective clearance.

## The impact of IL10 deficiency on the outcome of enteric protozoan infection

*Eimeria tenella* (*E. tenella*) is one of ten protozoan parasite species that can cause coccidiosis in chickens (*Blake et al., 2021*). *Eimeria* infection of chickens induces a robust IFNγ-driven immune response with evidence of higher IL10 expression associated with increased susceptibility (*Rothwell et al., 2004*; *Bremner et al., 2021*). Following an initial titration, an infectious dose of 7000 sporulated *E. tenella* oocysts per bird was set as the standard challenge, as this induced measurable phenotypes in pathology, performance, and parasite replication in the absence of overt disease in WT birds. Oral challenge with this or a mock dose was administered to cohorts of 10 IL10KO HOM and IL10EnKO HOM chickens and corresponding WT and HET controls on day 21 post-hatch. The outcomes are shown in *Figure 7*. Average parasite replication was determined by quantitative PCR (qPCR) as the ratio of parasite genomes detected per host genome in total caecal tissue collected 6 days post-infection. Parasite burden was almost ablated in IL10KO HOM and significantly reduced in IL10EnKO HOM chickens compared to their respective WT controls, whereas HET mutations had no significant effect in either case (*Figure 7A and B*). Histology confirmed reduced parasite replication in IL10KO HOM chickens, with dense populations of late-stage gametocytes observed in WT caecal tissues and correspondingly very sparse numbers in IL10KO HOM chickens (*Figure 7C and D*). This selective reduction of parasite burden in IL10KO HOM chickens was associated with a significant increase in caecal lesion score and a substantial reduction in body weight gain over the 6-day challenge interval (*Figure 7E and G*). By contrast, both IL10KO HET and IL10EnKO HOM and HET (each anticipated to reduce IL10 expression) showed reduced caecal lesion scores and had no effect on body weight gain compared to their respective WT controls (*Figure 7E–H*).

To better characterise the differences in pathology, performance and parasite replication detected between IL10KO WT and HOM chickens, we undertook a time course study, analysing parasite burden by qPCR at 0, 2, 4, 6, 8, and 10 days post-infection. In parallel, excreted oocysts per gram were counted in litter samples collected from IL10KO WT and HOM groups during the period of

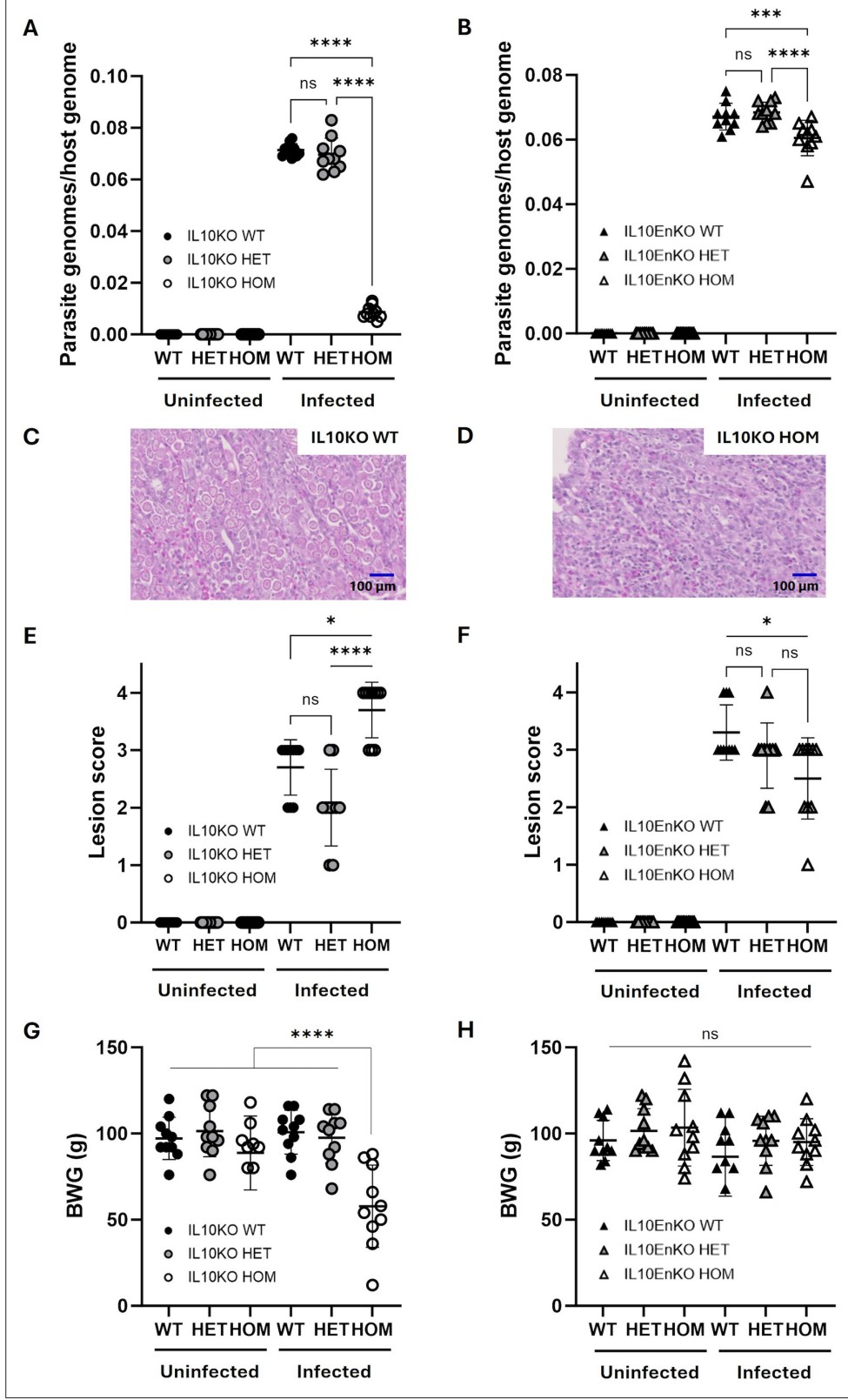

**Figure 7.** *E. tenella* replication and impact on host pathology and performance in wild-type (WT), heterozygous (HET), and homozygous (HOM) G2 populations of IL10 knockout (IL10KO) and IL10-Enhancer knockout (IL10EnKO) chickens. (**A, B**) Parasite replication calculated as parasite genomes per host genome detected by quantitative PCR in genomic DNA extracted from caecal tissue 6 days post-infection (dpi). (**C, D**) Haematoxylin and eosin

*Figure 7 continued on next page*

*Figure 7 continued*

stained caecal tissues collected from IL10KO WT and HOM chickens 6 days post *E. tenella* infection. Numerous gametocytes are visible in the IL10KO WT line (**C**); gametocytes are largely absent in the IL10KO HOM line (**D**). (**E, F**) Parasite-associated pathology measured as caecal lesion scores 6 dpi. (**G, H**) Body weight gain (BWG) over 6 days from time of challenge to sampling. All group sizes: n=10. Statistical significance calculated using one-way ANOVA with Tukey's multiple comparison tests (**A, B, G, H**) or Kruskal-Wallis with Dunn's multiple comparison tests (**E, F**); *p<0.05, ***p<0.001, ****p<0.0001, ns: not significant.

peak oocyst excretion at 7, 8, and 9 days post-infection. *Figure 8A* confirms the greatly reduced *E. tenella* replication in IL10KO HOM, which was reflected in corresponding reduced excretion of oocysts (*Figure 8B*). Lesions associated with *E. tenella* infection were accelerated in IL10KO HOM chickens compared to WT (*Figure 8C*). The adverse impact of IL10KO HOM on body weight gain was sustained to day 8 and showed signs of recovery by day 10 (*Figure 8D*), likely due to complete pathogen elimination (*Figure 8A*). Consistent with the mild inflammatory cell infiltration in IL10KO HOM birds noted in *Figure 7D*, inflammatory gene expression was elevated in the caecum prior to challenge (day 0). The greatest difference in IL10KO HOM birds relative to WT was detected on day 6, during the period of peak pathology (*Figure 8C and E*). Volcano plots showing the direction and magnitude of gene expression differences detected between IL10KO WT and HOM birds following *E. tenella* infection are shown in *Figure 8—figure supplement 1*, highlighting genes associated with inflammation (*Il6*) and lysosomes (*Hps5*). Transcription of genes associated with movement of molecules across membranes (*Pkd2l1*, *Abcg2*) and tubulin production (*Tubat*) was lower in IL10KO HOM birds 6 days post-infection.

## Discussion

An intrinsic compromise exists between effective host defence leading to pathogen elimination and the induction of immune-mediated pathology. The central importance of IL10 in mediating this balance has been studied extensively in the mouse, notably in the context of parasite infections (*Saraiva et al., 2020*; *Couper et al., 2008*; *Rasquinha et al., 2021*; *Piazzon et al., 2016*; *Redpath et al., 2014*). The regulatory function of IL10 is especially important in preventing inappropriate responses to the intestinal microbiome with the potential to drive spontaneous enterocolitis (*Kühn et al., 1993*; *Bernshtein et al., 2019*; *Zigmond et al., 2014*; *Büchler et al., 2012*). Here, we have generated two *IL10* mutations in the chicken genome, a definitive knockout and a putative enhancer mutation. Notwithstanding the development of robust technologies based upon CRISPR/Cas9, the development of models of immune deficiency in chickens remains time-consuming and logistically challenging, and relatively few informative mutations have been generated (*Wu et al., 2023*; *Heyl et al., 2023*). The enhancer mutation was intended to generate a quantitative impact on IL10 expression to model selection for reduced expression in production lines of birds. Based upon the mouse *Il10^{-/-}* phenotype, we also anticipated a possible severe post-hatch failure to thrive due to spontaneous colitis in IL10KO HOM chicks. An unexpected outcome was the discovery that the IL10KO in chicken is not dosage compensated; therefore, the HET provides a second model of the impact of reduced expression. In overview, neither the IL10KO HET nor the IL10EnKO HOM had a large effect on pathogen clearance or inflammation in either of the challenge models, indicating that IL10 biology is not highly sensitive to changes in expression of the magnitude that has been detected in individual lines of commercial broilers and layers (*Boulton et al., 2018a*; *Boulton et al., 2018b*; *Freem et al., 2019*).

Detailed analysis of the IL10KO HOM birds in SPF or conventional avian facilities revealed no evidence of spontaneous intestinal inflammation or compromised body weight gain, and only a small and inconsistent effect on specific antibody responses to routine live or inactivated vaccines. Since spontaneous colitis in mice depends upon resident enteric bacteria (*Sellon et al., 1998*), it is possible that the difference compared to the rodent models lies in the nature of the chicken intestinal microbiome (*Bajagai et al., 2024*), in which case the changes that we observed in the microbiome of IL10KO HOM chicks may serve to mitigate potential spontaneous intestinal pathology. In any case, the results indicate that deletion of IL10 from the genome or selection in favour of low IL10 expression alleles (*Boulton et al., 2018a*; *Ghebremicael et al., 2008*; *Boulton et al., 2018b*; *Freem et al., 2019*) need not necessarily affect production traits.

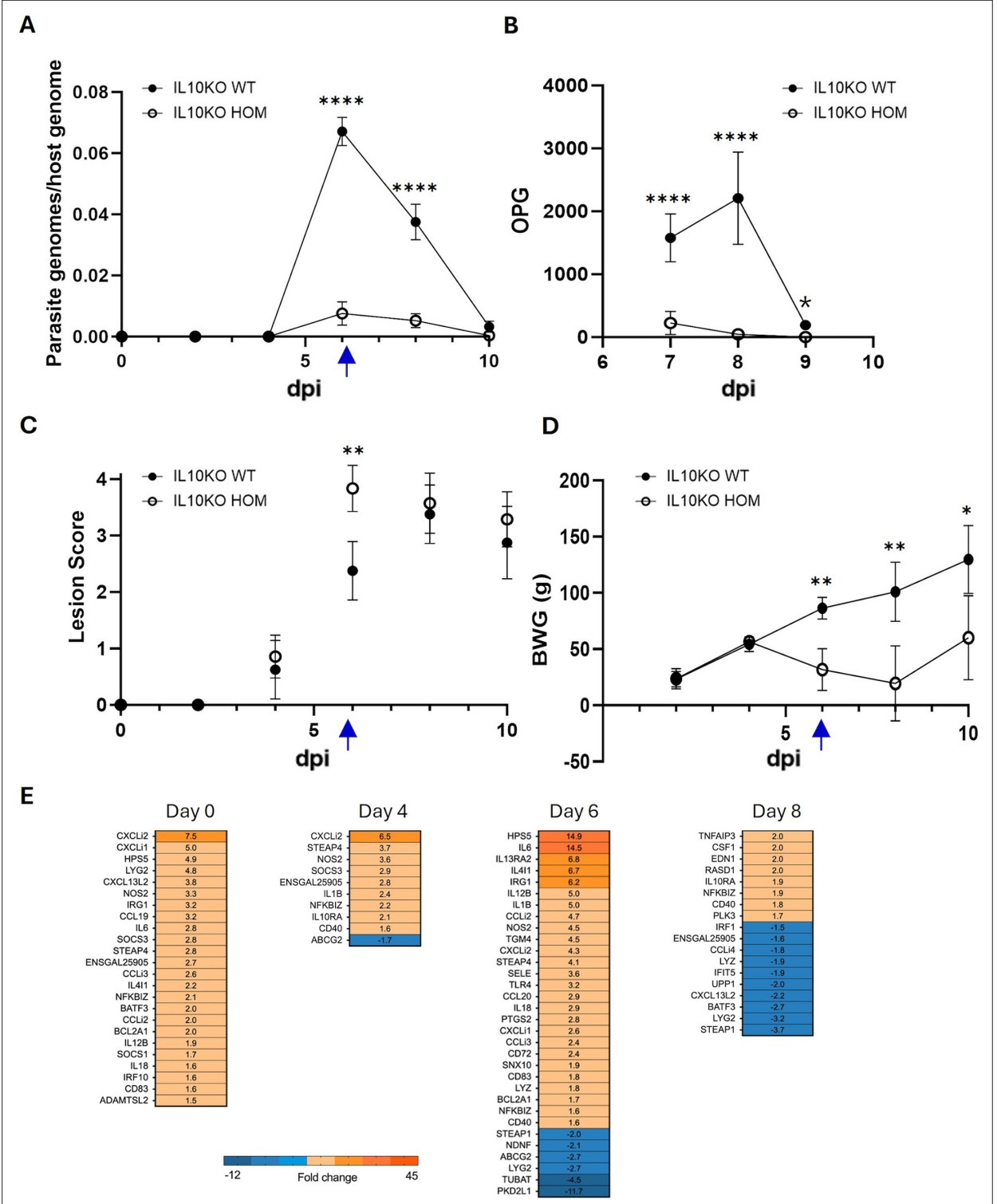

**Figure 8.** *E. tenella* infection time course in IL10 knockout (IL10KO) wild-type (WT) and homozygous (HOM) chickens. (**A**) Parasite replication calculated as parasite genomes per host genome detected by quantitative PCR in genomic DNA extracted from caecal tissue. (**B**) Oocysts per gram (OPG) litter. (**C**) Parasite-associated pathology measured as caecal lesion scores. (**D**) Body weight gain (BWG) over 10 days from time of challenge to final sampling. (**E**) Heat map of differentially transcribed genes in the IL10KO HOM caeca relative to IL10KO WT chickens 0, 4, 6, and 8 days post-infection (dpi). 48 chicks per group, eight culled from each group per timepoint. Statistical significance calculated using mixed-effects model (REML) with Tukey's multiple comparison tests (**A, D**), two-way ANOVA with Šídák's multiple comparison tests (**B**), or Mann-Whitney U tests (**C**); *p<0.05, **p<0.01, ***p<0.001, ****p<0.0001, ns: not significant. Blue arrows indicate the time of sampling in the comparative phenotyping study (*Figure 7*).

*Figure 8 continued on next page*

*Figure 8 continued*

The online version of this article includes the following figure supplement(s) for figure 8:

**Figure supplement 1.** Differential gene expression in the caecum of IL10 knockout (IL10KO) wild-type (WT) and homozygous (HOM) chickens infected with *E. tenella*.

In both the bacterial challenge models, and in the *Eimeria* challenge, the response of IL10KO HOM birds highlights for the first time in a non-rodent species the compromise between pathogen elimination and pathology. Accelerated pathogen clearance is associated with increased inflammatory gene expression and local inflammation, and in the case of *Eimeria*, reduced body weight gain. The models we have chosen are not lethal and there is effective pathogen clearance even in WT birds. Findings with *Campylobacter* in IL10KO HOM birds are consistent with elevated intestinal inflammation in a broiler breed with relatively low IL10 expression, albeit a significant reduction in caecal colonisation relative to breeds with higher IL10 expression was not observed (*Humphrey et al., 2014*). While the caecal load of *C. jejuni* was markedly reduced in IL10KO HOM birds at most intervals examined, some inconsistency existed in the timepoints when the reduction reached statistical significance between cohorts, likely as a consequence of inter-animal variation. This is typical of *C. jejuni* colonisation in chickens, where bacterial population structures have been reported to be variable and unpredictable (*Coward et al., 2008*). Similar variation between time intervals, birds and repeated experiments has been reported when evaluating vaccines against *C. jejuni* colonisation (e.g. *Buckley et al., 2010*; *Nothaft et al., 2021*). In mammals, *Salmonella* has been reported to gain a competitive advantage from intestinal inflammation, both owing to availability of tetrathionate that *Salmonella* can uniquely use as a respiratory electron acceptor (*Winter et al., 2010*) and remodelling of the indigenous microbiota (*Rogers, 2021*). It is unclear if *Salmonella* derives the same benefits during colonisation of the avian intestine and the counts of viable bacteria recovered from the caeca of WT and IL10KO HOM birds can only be interpreted as the balance of bacterial growth and killing.

Our results with *Eimeria* are relevant to commercial studies aimed at targeting IL10 to mitigate disease. Coccidiosis has been estimated to exert costs in excess of £10 billion every year to the chicken production industry (*Blake et al., 2020*), with *E. tenella* among the most common and pathogenic species (*Clark et al., 2016*; *Long et al., 1976*). Feeding neutralising anti-IL10 antibody to chickens during *Eimeria* infection was reported to mitigate reduced body weight gain (BWG) but had an inconsistent impact on parasite replication and pathology (*Rasheed et al., 2020*; *Sand et al., 2016*). Similarly, feeding antibody to IL10 receptor 2 (but not receptor 1) reduced lost BWG, but had no effect on oocyst shedding (*Arendt et al., 2019*). Feeding chickens genetically modified corn (*Zea mays*) expressing IL10 improved BWG and reduced oocyst shedding and caecal lesions (*Lessard et al., 2020*). These results are consistent with the phenotypes associated with a reduction, rather than complete absence, of IL10 observed here.

Given the excessive inflammatory activation we observed in non-lethal infections, and the literature in mice, IL10KO HOM birds or hypomorphs might be hyper-sensitive to more virulent bacterial infections and viral pathogens, such as avian influenza, that can elicit a cytokine storm. On the other hand, they may be primed for a more rapid and effective response. These possibilities can now be tested using the models we have developed.

## Materials and methods
### Animals
All chicken lines used in this study were bred and maintained at the NARF, University of Edinburgh, under UK Home Office Establishment Licence and in compliance with the Animals (Scientific) Procedures Act (ASPA) 1986. All regulated procedures were approved by local ethical review committees and carried out by UK Home Office Personal Licence holders under UK Home Office Project Licences 70/8528 and PP9565661 (creation and maintenance of genetically altered chicken lines), PCD70CB48 (*Campylobacter* and *Salmonella* challenges performed at the Moredun Research Institute), and PDAAD5C9D (*Eimeria* challenges performed at the Royal Veterinary College). Animals were humanely culled in accordance with Schedule 1, ASPA 1986.

The RIR line was maintained as a closed outbred population in the NARF SPF avian facility. The CSF1R-eGFP line (*Balic et al., 2014*) was created and maintained on a Hy-Line background in the

NARF conventional avian facility by HOM x HOM crosses; CSF1R-eGFP fertile eggs were used for PGC derivation. The iCaspase9 line (*Ballantyne et al., 2021b*) was created and maintained on a RIR background in the NARF SPF avian facility by crossing iCaspase9 HET cockerels to RIR hens; iCaspase9 fertile eggs were used as hosts for PGC injection. The IL10KO and IL10EnKO lines were created in the NARF SPF avian facility and maintained on a mixed Hy-Line/RIR background by crossing iCaspase9 surrogate host cockerels carrying gene-edited PGCs (generation G0) to RIR hens to produce the first generation (G1) of IL10KO HET and IL10EnKO HET chickens (see *Supplementary file 1* for detailed G1 bird numbers); the second generation (G2) of IL10KO and IL10EnKO chickens was obtained by crossing G1 HET cockerels to G1 HET hens from each line, resulting in WT, HET, and HOM offspring (see *Supplementary file 1* for detailed G2 bird numbers).

## Chicken PGC derivation and culture

PGC lines were derived from blood collected from CSF1R-eGFP HOM embryos at Hamburger-Hamilton (HH) stage 16 and expanded in FAOT medium as previously described (*Whyte et al., 2015*). A W-chromosome-specific PCR (*Clinton, 1994*) was used to determine the sex of the PGC lines. PGC lines were tested for *Mycoplasma* contamination by PCR using primers specific for *Mycoplasma synoviae* (MS; Myco5_14,713A and Myco3_14709) and *Mycoplasma gallisepticum* (MG; Myco5_14712 and Myco3_14709); see *Supplementary file 3* for primer sequences.

## CRISPR plasmid, guide RNAs, and ssODN donor

Guide RNAs (gRNAs) were designed using CHOPCHOP (*Labun et al., 2019*) and CRISPOR (*Concordet and Haeussler, 2018*) and cloned into the High Fidelity CRISPR/Cas9 plasmid HF-PX459 V2.0 (*Idoko-Akoh et al., 2018*) as described previously (*Ran et al., 2013*). A gRNA (ggIL10_exon1_g3) and a 143 bp single-stranded oligonucleotide DNA (ssODN; IL10_exon1_HDRoligo2; IDT Ultramer DNA Oligonucleotide) containing a STOP codon (TAG) and AvrII restriction site (CCTAGG) were used to edit *IL10* exon 1. Two gRNAs (IL10_enhancer_guide1 and IL10_enhancer_guide2) were used in combination to delete the *IL10* putative enhancer region. See *Supplementary file 4* for gRNA and ssODN sequences.

## PGC transfection and screening

To edit *IL10* exon 1, 200,000 CSF1R-eGFP male PGCs were transiently transfected with 1.5 µg gRNA plasmid ggIL10_exon1_g3 and 10 pmol ssODN IL10_exon1_HDRoligo2. To delete the *IL10* putative enhancer region, 200,000 CSF1R-eGFP male PGCs were transiently transfected with 1.0 µg gRNA plasmid IL10_enhancer_guide1 and 1.0 µg gRNA plasmid IL10_enhancer_guide2. Transient transfections were performed with Lipofectamine 2000 (Invitrogen), followed by treatment with 0.4 µg/mL puromycin to enrich for transfected PGCs, as previously described (*Idoko-Akoh et al., 2018*). Bulk PGC populations were screened by PCR to confirm the presence of the desired IL10KO or IL10EnKO edits. Single-cell clonal populations were subsequently expanded as previously described (*Idoko-Akoh et al., 2018*) and screened by PCR and sequencing. Primers IL10_Exon1_F4 and IL10_Exon1_R4 were used to amplify a 1826 bp fragment encompassing *IL10* exon 1, followed by digestion with AvrII to identify PGCs carrying the IL10KO-edited allele (*Figure 1—figure supplement 4A*); primers IL10_Exon1_F1 and IL10_Exon1_R1 were subsequently used to sequence the *IL10* exon 1 fragment. Primers IL10-Enhancer_F2 and IL10_Enhancer_R2 were used to amplify and sequence a fragment encompassing the *IL10* putative enhancer region (*Figure 1—figure supplement 4C*). See *Supplementary file 3* for primer sequences and expected fragment sizes. PGCs were transfected and cultured under SPF conditions and regularly tested for mycoplasma contamination, as described above.

## Generation of surrogate host chickens and establishment of the IL10KO and IL10EnKO lines under SPF conditions

Male PGCs carrying the desired IL10KO or IL10EnKO biallelic edits were injected into iCaspase9 embryos at HH stage 16, together with the iCaspase 9 activator AP20187 (Takara Bio) as previously described (*Ballantyne et al., 2021b*), to create surrogate host cockerels carrying IL10KO HOM or IL10EnKO HOM germ cells (G0 founders); all manipulations were performed under SPF conditions in the NARF SPF avian facility. G0 founders were hatched and raised to sexual maturity in negative pressure isolators and released to floor pens at 22 weeks. The SPF status of the G0 founders was

externally assessed at three timepoints (10, 20, and 30 weeks post-hatch) by screening for 17 different pathogens (Sci-Tech Ireland; *Supplementary file 5*).

G0 founder cockerels were bred to RIR hens to produce the first generation (G1) of IL10KO HET and IL10EnKO HET chickens (see *Supplementary file 1* for detailed G1 bird numbers); the second generation (G2) of IL10KO and IL10EnKO chickens was obtained by crossing G1 HET cockerels to G1 HET hens, resulting in IL10KO and IL10EnKO WT, HET, and HOM offspring (see *Supplementary file 1* for detailed G2 bird numbers). All G1 and G2 chicks were genotyped by PCR using genomic DNA extracted from chorioallantoic membrane or 4 µL blood samples as previously described (*Ballantyne et al., 2021b*). Primers IL10_Exon1_F1 and IL10_Exon1_R1 were used to amplify a 246 bp fragment encompassing *IL10* exon 1, followed by digestion with AvrII to identify the IL10KO WT and edited alleles (*Figure 1—figure supplement 4B*). Primers IL10-Enhancer_F2 and IL10_Enhancer_R2 were used to amplify the IL10EnKO WT and edited alleles (*Figure 1—figure supplement 4D*). Primers LTR_U3_F, 235_F and 235_R were used to amplify the CSF1R-eGFP WT and transgenic alleles. See *Supplementary file 3* for primer sequences and expected fragment sizes. The SPF status of G1 and G2 chickens was verified at point of lay (20–24 weeks) and at end of life (around 40 weeks).

## Animal monitoring, post-mortem examination, and histopathological analyses under SPF and conventional conditions

All IL10KO and IL10EnKO chicks hatched in the NARF SPF avian facility were closely monitored for signs of reduced growth, health issues, adverse phenotypes, or unexpected behaviours. All chicks were weighed daily in the first week post-hatch and twice a week until 4 weeks; a small cohort of G2 IL10KO and IL10EnKO chicks was subsequently weighed weekly, up to a maximum of 30 weeks. Post-mortem examination of one WT and one HOM bird from both IL10KO and IL10EnKO lines was performed monthly from 16 to 40 weeks, with a particular focus on the gastrointestinal tract. Tissue samples from the proximal colon, ileum, and duodenum were collected for histopathological analyses at all timepoints and scored in a blinded manner using a semiquantitative scoring system (*McCafferty et al., 2000*) to grade the extent of destruction of normal mucosal architecture (0–3), the presence and degree of cellular infiltration (0–3), the extent of muscle thickening (0–3), the presence or absence of crypt abscesses (0–1), and the presence or absence of goblet cell depletion (0–1), for a maximum possible score of 11.

In addition to animals raised in the NARF SPF avian facility, a single cohort of IL10KO chicks was hatched in the NARF conventional avian facility and closely monitored as described above. Post-mortem examination of one WT and one HOM IL10KO bird was performed monthly from 10 to 50 weeks, with a particular focus on the gastrointestinal tract. Tissue samples from the proximal colon, duodenum, and caecum were collected for histopathological analyses and scored as described above.

## Generation of BMDMs from chicken embryos

Bone marrow cells were isolated from the tibias and femurs of day 18 embryos (by pooling three embryos of the same genotype per sample) by flushing the bone marrow through a 40 µm cell strainer with RPMI-1640 medium (Sigma R5886) supplemented with 10% fetal bovine serum ultra-low IgG (Gibco 011-90035M), 2 mM L-glutamine (Gibco 25030081), and 0.1X penicillin/streptomycin (Gibco 15140122) using a syringe and blunt needle. Bone marrow cells were washed, resuspended at $1\times10^6$ cells/mL in supplemented RPMI-1640 medium, seeded at 1 mL per well in six-well plates or 150 µL per well in 96-well plates, and cultured for 7 days at 41°C, 5% $CO_2$ in the presence of 200 ng/mL colony stimulating factor-1 to induce formation of BMDMs. Cell culture supernatants were subsequently used for capture ELISA and nitric oxide assay, as detailed below.

## Detection of IL10 by capture ELISA

On day 7 of culture, BMDMs cultured in six-well plates were stimulated with 0.5 µg/mL LPS (from *E. coli* O55:B5; Sigma L2880) for 2 hr at 41°C, 5% $CO_2$ to induce IL10 expression; supernatants were then harvested, and IL10 protein levels were measured by capture ELISA, as described previously (*Wu et al., 2016*). Briefly, assay plates were coated with 3 µg/mL capture antibody (ROS-AV164; *Wu et al., 2016*) incubated overnight at 4°C, washed and blocked. Plates were then incubated with recombinant IL10 standards or 50 µL of 10-fold diluted cell culture supernatants, followed by incubation with 1 µg/mL biotinylated detection antibody (ROS-AV163; *Wu et al., 2016*) and then incubation with Pierce

High Sensitivity Streptavidin-HRP (1:5000; Thermo Fisher Scientific, cat #34028), before adding 1-Step TMB ELISA Substrate Solution (Thermo Fisher Scientific, cat #21130) and then sulfuric acid stop solution. Absorbance was measured at 450 nm (550 nm as a reference) in a SpectraMax 250 microplate spectrophotometer system (Molecular Devices, Sunnyvale, CA, USA). Data were fitted with GraphPad Prism 7.0 using a second-order polynomial (quadratic) model; statistical significance was calculated using two-tailed unpaired t tests.

### Nitric oxide assay

On day 7 of culture, BMDMs cultured in 96-well plates were stimulated with 0.5 µg/mL LPS in the presence or absence of 2.5 µg/mL neutralising anti-IL10 antibody (ROS-AV163; *Wu et al., 2016*) for 24 hr at 41°C, 5% $CO_2$; supernatants were then harvested and nitrite levels were measured by Griess assay, according to the manufacturer's instructions (Promega G2930); absorbance was measured at 550 nm. Statistical significance was calculated using two-tailed unpaired t tests.

### Response to vaccination

IL10KO WT and HOM chicks hatched and raised in the NARF conventional avian facility were vaccinated following the routine vaccination schedule in place in the facility (*Supplementary file 2*). One mL blood samples were collected from the brachial vein of 29-week-old hens and sent fresh to Sci-Tech Ireland for antibody titre measurements by ELISA. Statistical significance was calculated using Mann-Whitney U tests.

### Microbiome analysis

Caecal contents were harvested from 4-week-old IL10KO WT and HOM chicks raised in the NARF SPF avian facility. DNA isolation, amplification of the V3-V4 region of the 16S rDNA gene, and paired-end sequencing were performed as described previously (*Salavati Schmitz et al., 2024*). Raw reads were quality assessed using fastQC, and the DADA2 plugin in QIIME2 was used to generate an ASVs table against the SILVA138 database. Alpha diversity and weighted uniFrac distances were calculated using the Phyloseq package, while Bray-Curtis metrics were generated using the vegan package. PERMANOVA (Bray-Curtis) analysis was conducted to test for significant effects of the group on overall microbiome community composition. LEfSe analysis was performed to identify differentially abundant phyla and genera between both groups using an LDA score equal to 4 as a threshold value. p-Values were declared significant at p<0.05. The sequencing data have been deposited in the European Nucleotide Archive (ENA) at EMBL-EBI under accession number PRJEB71960.

### Bacterial strains

*C. jejuni* strain M1 was used, as the minimum number of viable bacteria required for reliable intestinal colonisation of chickens has been defined (*Vohra et al., 2020*). Strain M1 was grown on charcoal cefoperazone deoxycholate agar (CCDA) at 41°C for 48 hr in a microaerophilic atmosphere (90 % $N_2$, 5% $CO_2$, and 5% $O_2$). For the challenge, 10 mL of Mueller Hinton broth was inoculated with single colonies and incubated for 16 hr with shaking at 41°C in the same atmosphere. Prior to inoculation, bacteria were confirmed to have spiral morphology and to be motile by phase contrast microscopy. Cultures were diluted in phosphate-buffered saline (PBS) based on a standard curve of colony-forming units (CFU/mL) relative to optical density at 600 nm, and inocula were retrospectively confirmed by plating serial ten-fold dilutions onto CCDA. A spontaneous nalidixic acid-resistant mutant of *Salmonella enterica* serovar Typhimurium strain 4/74 was used, as its colonisation kinetics are well defined in chickens (*Chaudhuri et al., 2013*). This was cultured on Luria-Bertani (LB) agar containing 20 µg/mL nalidixic acid at 37°C overnight, and a single colony was transferred to LB broth with nalidixic acid and incubated at 37°C with shaking for 20 hr. 100 µL of this culture was used for inoculation, and the number of viable bacteria given was determined retrospectively by plating serial ten-fold dilutions onto LB agar.

### Experimental infection of chickens

Animal experiments using *Campylobacter* or *Salmonella* were conducted at the Moredun Research Institute. Birds were provided with access to water and irradiated feed based on vegetable protein *ad libitum*. Groups of 20–22 birds of each genotype (IL10KO WT, HET, HOM) were wing-tagged for

identification and separately housed in biosafety level 2 facilities from day 7 of age in colony cages. At 14 days of age, birds were inoculated by oral gavage with 100 µL of bacterial suspension. In two separate trials with *C. jejuni*, birds were inoculated with $1 \times 10^2$ CFU of strain M1. In a single trial with *S.* Typhimurium, birds were inoculated with $1 \times 10^8$ CFU of strain 4/74 nal$^R$. At 7 days post-inoculation, half of the birds in each group were euthanised by cervical dislocation. The remainder of the birds was euthanised at 14 days post-inoculation. On post-mortem examination, samples from the ileum and the distal end of the caecum were stored in RNA*later* or fixed in 10% neutral-buffered formalin for histopathology. The contents of the caeca were pooled at the bird level, homogenised in PBS, and ten-fold serial dilutions cultured on CCDA at 41°C in a microaerophilic atmosphere to enumerate CFU/g of *C. jejuni* or MacConkey agar containing 20 µg/mL nalidixic acid at 37°C aerobically to enumerate CFU/g of *Salmonella*.

## Histopathology

Formalin-fixed tissue samples were embedded in paraffin, sectioned, mounted on glass slides, and stained by haematoxylin-eosin (HE) staining. Histopathological analyses were performed by a certified veterinary pathologist blind to the genotype of individuals and the inoculum. Samples of ileum and caecum were assessed for the presence of lymphoplasmacytic and heterophil infiltration on a scale of 0–5 (where infiltration is 0 normal, 1 slight, 2 mild, 3 moderate, 4 marked, or 5 severe).

## RNA extraction and multiplex PCR

Caecal tissue samples in RNA*later* were homogenised with a hand-held tissue homogeniser and RNA was extracted using the QIAGEN RNeasy kit following the manufacturer's instructions. The concentration and integrity of the RNA were respectively analysed using a NanoDrop microvolume spectrophotometer and TapeStation automated electrophoresis system. Reverse transcription, preamplification, and high-throughput qPCR for avian immune-related transcripts and reference genes were performed as previously described using a 96.96 Dynamic Array IFC for Gene Expression (*Borowska et al., 2019*; Standard BioTools, San Francisco, CA, USA).

## Parasites

Animal experiments using *Eimeria* were conducted at the Royal Veterinary College, using chicks hatched at the NARF SPF avian facility. All chickens were provided with access to water and ammonia fumigated feed based on vegetable protein *ad libitum*. The *E. tenella* Houghton (H) reference strain was used (*Aunin et al., 2021*). Parasites were passaged by oral inoculation of SPF Lohmann Valo chickens, purified and prepared for use as described elsewhere (*Long et al., 1976*). At 21 days post-hatch, chicks were weighed and infected by oral inoculation with 7000 sporulated *E. tenella* oocysts or mock-infected using sterile water. Group sizes (minimum eight to ten per group) were determined following dose titration in IL10KO and IL10EnKO HET chickens using power calculations with alpha set at 0.05 and power at 80%. At the desired timepoint, birds were culled and whole blood was collected for serum. Caeca were collected, scored for severity of infection (*Johnson and Reid, 1970*), and preserved using RNA*later* (Invitrogen). Total genomic DNA (gDNA) was extracted from thawed complete caeca stored in RNA*later* from all studies. Caeca were homogenised in Buffer ATL using a TissueRuptor homogeniser (QIAGEN, Hilden, Germany) and then digested overnight at 56°C in Buffer ATL and proteinase K, prior to extraction using the QIAGEN DNeasy Blood and Tissue DNA Kit according to the manufacturer's instructions.

qPCR for assessment of *E. tenella* genome copy number was performed as previously described (*Nolan et al., 2015*). Briefly, gDNA extracted from caecal tissue was used as template for qPCR targeting *E. tenella* (RAPD-SCAR marker Tn-E03-116, primers F: 5'-TCGTCTTTGGCTGGCTATTC-3', R: 5'-CAGAGAGTCGCCGTCACAGT-3'), normalised against the number of chicken tata-binding protein (TBP) genomic copies detected (F: 5'-TAGCCCGATGATGCCGTAT-3', R: 5'-GTTCCCTGTGTCGCTT GC-3'). qPCR was performed in 20 µL reactions in triplicate containing 10 µL 2×SsoFast EvaGreen Supermix (Bio-Rad, Hercules, CA, USA), 1 µL of primers (F: 3 µM, R: 3 µM), 8 µL of molecular biological grade water (Invitrogen) and 1 µL of gDNA, or water as negative control. Hard-shelled 96-well reaction plates (Bio-Rad) were sealed with adhesive film (Bio-Rad) and loaded into a Bio-Rad CFX qPCR cycler. Reactions were heated to 95°C for 2 min, prior to 40 cycles consisting of 95°C for 15 s then 60°C for 30 s with a fluorescence reading taken after each cycle. Melting curve analysis was performed for 15 s

at 95°C, before cooling to 65°C for 60 s, then heating to 95°C in 0.5°C increments for 0.5 s. Absolute quantification was performed against a standard curve generated using serially diluted plasmid DNA containing the amplicon of interest (EtenSCAR or ChickenTBP), to generate a standard curve ranging from $10^6$ to $10^1$ genome copies per mL. Parasite genome copy number was normalised by division with host (chicken) genome copy number.

## Statistical analyses

Minitab Statistical Software was used to analyse IL10 protein levels and nitric oxide production in BMDMs using two-tailed unpaired t tests; growth curves were analysed using two-tailed unpaired t tests or one-way ANOVA with Bonferroni multiple comparison tests; histopathological scores were analysed using Mann-Whitney U tests or Kruskal-Wallis tests. Analysis of bacterial counts, histopathology, and heat maps was performed with GraphPad Prism software version 10.2.0. Data from multiplex RT-qPCR was analysed with GenEx6 software. The distribution of data was analysed by D'Agostino-Pearson and Shapiro-Wilk normality tests. *Eimeria* replication and chicken BWG during parasite infection were analysed using one-way ANOVA with Tukey's multiple comparison tests, and differences in lesion scores were tested by Kruskal-Wallis with Dunn's multiple comparison tests. Bacterial counts and histopathology scores were analysed by Mann-Whitney U tests or Kruskal-Wallis tests. Fold change in transcript abundance was analysed by t tests using normalised data.

## Acknowledgements

We thank staff at the National Avian Research Facility of the Roslin Institute, at the Moredun Research Institute, and at the Biological Services Unit of the Royal Veterinary College, for care and maintenance of the chickens used in this study.

## Additional information

### Competing interests

Michael J McGrew: is inventor on patent application WO 2020074915 for the iCaspase9 surrogate host chicken; the University of Edinburgh is the applicant. The other authors declare that no competing interests exist.

### Funding

| Funder | Grant reference number | Author |
| --- | --- | --- |
| Biotechnology and Biological Sciences Research Council | BB/P022049/1 | Kay Boulton<br>Androniki Psifidi<br>Kellie A Watson<br>Michael J McGrew<br>Mark P Stevens<br>David A Hume |
| Biotechnology and Biological Sciences Research Council | BB/P021638/1 | Fiona Tomley<br>Damer P Blake |
| Cobb-Vantress | 13393976 | Dominique Meunier<br>Kellie A Watson<br>Michael J McGrew |
| Roslin Institute | 13302777 | Dominique Meunier |

| Funder | Grant reference number | Author |
|---|---|---|
| Biotechnology and Biological Sciences Research Council | BBS/E/D/20002174 | Dominique Meunier<br>Ricardo Corona-Torres<br>Kay Boulton<br>Zhiguang Wu<br>Maeve Ballantyne<br>Laura Glendinning<br>Anum Ali Ahmad<br>Dominika Borowska<br>Lorna Taylor<br>Lonneke Vervelde<br>Kellie A Watson<br>Michael J McGrew<br>Mark P Stevens |
| Biotechnology and Biological Sciences Research Council | BBS/E/RL/230002B | Dominique Meunier<br>Ricardo Corona-Torres<br>Kay Boulton<br>Zhiguang Wu<br>Maeve Ballantyne<br>Laura Glendinning<br>Anum Ali Ahmad<br>Dominika Borowska<br>Lorna Taylor<br>Lonneke Vervelde<br>Kellie A Watson<br>Michael J McGrew<br>Mark P Stevens |

The funders had no role in study design, data collection and interpretation, or the decision to submit the work for publication.

## Author contributions

Dominique Meunier, Conceptualization, Formal analysis, Funding acquisition, Investigation, Methodology, Validation, Visualization, Writing – original draft, Writing – review and editing; Ricardo Corona-Torres, Formal analysis, Investigation, Methodology, Validation, Visualization, Writing – original draft, Writing – review and editing; Kay Boulton, Conceptualization, Funding acquisition, Investigation, Writing – review and editing; Zhiguang Wu, Formal analysis, Investigation, Methodology, Resources, Validation, Writing – review and editing; Maeve Ballantyne, Investigation, Methodology; Laura Glendinning, Formal analysis, Investigation, Methodology, Resources, Supervision, Writing – review and editing; Anum Ali Ahmad, Formal analysis, Investigation, Visualization, Validation; Dominika Borowska, Formal analysis, Writing – review and editing; Lorna Taylor, José Jaramillo-Ortiz, Investigation; Lonneke Vervelde, Formal analysis, Resources, Supervision, Writing – review and editing; Jorge del Pozo, Formal analysis; Marili Vasilogianni, Gonzalo Sanchez-Arsuaga, Investigation, Writing – review and editing; Androniki Psifidi, Conceptualization, Funding acquisition, Writing – review and editing; Fiona Tomley, Conceptualization, Funding acquisition, Supervision, Writing – review and editing; Kellie A Watson, Conceptualization, Funding acquisition, Supervision; Michael J McGrew, Conceptualization, Formal analysis, Funding acquisition, Methodology, Resources, Supervision, Writing – review and editing; Mark P Stevens, Conceptualization, Formal analysis, Funding acquisition, Methodology, Resources, Supervision, Writing – original draft, Writing – review and editing; Damer P Blake, Conceptualization, Formal analysis, Funding acquisition, Investigation, Methodology, Resources, Supervision, Validation, Visualization, Writing – original draft, Writing – review and editing; David A Hume, Conceptualization, Formal analysis, Funding acquisition, Investigation, Supervision, Writing – original draft, Writing – review and editing

## Author ORCIDs

Dominique Meunier ⓘ https://orcid.org/0000-0001-7208-4221
Ricardo Corona-Torres ⓘ https://orcid.org/0000-0003-1492-1997
Zhiguang Wu ⓘ https://orcid.org/0000-0001-9131-730X
Laura Glendinning ⓘ https://orcid.org/0000-0003-4789-6644
Marili Vasilogianni ⓘ https://orcid.org/0009-0005-1940-3891
Michael J McGrew ⓘ https://orcid.org/0000-0001-8213-4632
Damer P Blake ⓘ https://orcid.org/0000-0003-1077-2306

David A Hume [iD] https://orcid.org/0000-0002-2615-1478

### Ethics

All chicken lines used in this study were bred and maintained at the National Avian Research Facility (NARF), University of Edinburgh, under UK Home Office Establishment Licence and in compliance with the Animals (Scientific) Procedures Act (ASPA) 1986. All regulated procedures were approved by local ethical review committees and carried out by UK Home Office Personal Licence holders under UK Home Office Project Licences 70/8528 and PP9565661 (creation and maintenance of genetically altered chicken lines), PDAAD5C9D (Eimeria infection) and PCD70CB48 (Campylobacter and Salmonella infection). Animals were humanely culled in accordance with Schedule 1, ASPA 1986.

Reviewer #1 (Public review): https://doi.org/10.7554/eLife.106252.3.sa1
Reviewer #2 (Public review): https://doi.org/10.7554/eLife.106252.3.sa2
Author response https://doi.org/10.7554/eLife.106252.3.sa3

---

## Additional files

### Supplementary files

Supplementary file 1. Number of IL10 knockout (IL10KO) and IL10-Enhancer knockout (IL10EnKO) wild-type (WT), heterozygous (HET), and homozygous (HOM) chicks hatched in the National Avian Research Facility (NARF) specified pathogen-free (SPF) chicken facility in the first (G1) and second (G2) generations. n.d.: not determined.

Supplementary file 2. Routine vaccination schedule in the National Avian Research Facility (NARF) conventional chicken facility.

Supplementary file 3. PCR primer sequences and expected fragment sizes.

Supplementary file 4. Guide RNA and ssODN sequences. Note the three substituted nucleotides (red, lowercase) and AvrII restriction site (CCTAGG, underlined) in the ssODN sequence.

Supplementary file 5. National Avian Research Facility (NARF) specified pathogen-free (SPF) screening.

MDAR checklist

### Data availability

All data generated or analysed during the study are included in the article, figures, figure supplements or supplementary files; microbiome sequencing data have been deposited in the European Nucleotide Archive (ENA) at EMBL-EBI under accession number PRJEB71960. IL10KO- and IL10EnKO-edited PGCs (HET and HOM) are cryopreserved at the Roslin Institute, University of Edinburgh, and available to researchers under MTA.

The following dataset was generated:

| Author(s) | Year | Dataset title | Dataset URL | Database and Identifier |
|---|---|---|---|---|
| Glendinning L | 2024 | 16S analysis of caecal contents of IL10 chicken | https://www.ebi.ac.uk/ena/browser/view/PRJEB71960 | EBI European Nucleotide Archive, PRJEB71960 |

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
