## [Editor Report · eLife Assessment]

IL10 balances protective and deleterious immune functions in mice and humans, but if IL10 also controls avian intestinal homeostasis remains unclear. Generating genetic knockouts, Meunier et al. established that a complete lack of IL10 strengthened immunity against enteric bacteria in chickens, while also aggravating infection-inflicted inflammatory tissue damage and dysbiosis upon parasite infection, but unlike mouse models, IL10 deficiency did not provoke spontaneous colitis in chickens. The findings presented are **valuable**, and the strength of evidence is **convincing**. The observation may have implications for the livestock industry and additional studies involving genetic knockouts may further unravel conserved and distinct avian IL10 controls.

---

## [Referee Report · Reviewer #1 (Public review)]

Summary:

In this study, Meunier et al. investigated the functional role of IL-10 in avian mucosal immunity. While the anti-inflammatory role of IL-10 is well established in mammals, and several confirmatory Knock-out models available in mice, IL-10's role in avian mucosal immunity is so far correlative. In this study the authors generated two different models of IL-10 ablation in Chickens. A whole body knock-out model, and an enhancer KO model leading to reduced IL10 expression. The authors first performed in vitro LPS stimulation based experiments, and then in vivo two different infection models employing C. jejuni and E. tenella, to demonstrate that complete ablation of IL10 leads to enhanced inflammation related pathology and gene expression, and enhanced pathogen clearance. At a steady-state level, however, IL-10 ablation did not lead to spontaneous colitis.

Strengths:

Overall the study is well executed and establishes an anti-inflammatory role of IL-10 in birds. While the results are expected, and not surprising, this appears to be the first report to conclusively demonstrate IL-10's anti-inflammatory role upon its genetic ablation in avian model. Provided the applicability of this information in combating pathogen infection in livestock species in sustainable industries like poultry, the study is worth publishing.

Weaknesses:

The study is primarily a confirmation of the already established anti-inflammatory role of IL-10.

Comments on revised version:

The authors have incorporated most of the points raised, and provided a reasonable argument for not considering DSS mediated colitis as an additional model.

---

## [Referee Report · Reviewer #2 (Public review)]

Summary:

The authors were to investigate functional role of IL10 on mucosal immunity in chickens. CRISPR technology was employed to generate IL10 knock out chickens in both exon and putative enhancer regions. IL10 expressions were either abolished (knockout in exon) or reduced (enhancer knock-out). IL-10 play an important role in the composition of the caecal microbiome. Through various enteric pathogens challenge, deficient IL10 expression was associated with enhanced pathogen clearance, but with more severe lesion score and body weight loss.

Strengths:

Both in vitro and in vivo knock-out in abolished and reduced IL10 expression and broad enteric pathogens were challenged in vivo and various parameters were examined to evaluate the functional role of IL10 on mucosal immunity.

Weaknesses:

Overexpression of IL10 either in vitro or in vivo may further support the findings from this study.

Comments on revised version:

The authors' response and justifications are appropriate.

---

## [Author Response]

The following is the authors’ response to the original reviews.

**Reviewer #1 (Public review):**
Summary:In this study, Meunier et al. investigated the functional role of IL-10 in avian mucosal immunity. While the anti-inflammatory role of IL-10 is well established in mammals, and several confirmatory knockout models are available in mice, IL-10's role in avian mucosal immunity is so far correlative. In this study, the authors generated two different models of IL-10 ablation in Chickens. A whole body knock-out model and an enhancer KO model leading to reduced IL10 expression. The authors first performed in vitro LPS stimulation-based experiments, and then in vivo two different infection models employing C. jejuni and E. tenella, to demonstrate that complete ablation of IL10 leads to enhanced inflammation-related pathology and gene expression, and enhanced pathogen clearance. At a steady-state level, however, IL-10 ablation did not lead to spontaneous colitis.Strengths:Overall, the study is well executed and establishes an anti-inflammatory role of IL-10 in birds. While the results are expected and not surprising, this appears to be the first report to conclusively demonstrate IL-10's anti-inflammatory role upon its genetic ablation in the avian model. Provided this information is applicable in combating pathogen infection in livestock species in sustainable industries like poultry, the study will be of interest to the field.Weaknesses:The study is primarily a confirmation of the already established anti-inflammatory role of IL-10.

We do not agree that this work is primarily confirmatory. The anti-inflammatory role of IL10 was indeed known previously from studies in mammals. The much more general insight from the current study is our demonstration of the intrinsic trade-off between inflammation and tolerance in the response to both the microbiome (which was significantly altered in the IL10 knockout birds) and mucosal pathogens. The study of *Eimeria* challenge in particular highlights the fact that it may be better for the host to tolerate a potential pathogen than to take on the cost of elimination.

**Reviewer #2 (Public review):**
Summary:The authors were to investigate the functional role of IL10 on mucosal immunity in chickens. CRISPR technology was employed to generate IL10 knock-out chickens in both exon and putative enhancer regions. IL10 expressions were either abolished (knockout in exon) or reduced (enhancer knock-out). IL-10 plays an important role in the composition of the caecal microbiome. Through various enteric pathogen challenges, deficient IL10 expression was associated with enhanced pathogen clearance, but with more severe lesion scores and body weight loss.Strengths:Both in vitro and in vivo knock-out abolished and reduced IL10 expression, and broad enteric pathogens were challenged in vivo, and various parameters were examined to evaluate the functional role of IL10 on mucosal immunity.Weaknesses:Overexpression of IL-10 either in vitro or in vivo may further support the findings from this study.

An overexpression experiment, regardless of outcome, would not necessarily support or invalidate the findings of the current study. It would address the question of whether the absolute concentration of IL10 produced alters the outcome of an infection.

**Reviewer #1 (Recommendations for the authors):**
The following are the recommendations that, in my opinion, will be helpful to enhance the quality of the study.Major point:The authors at a steady state did not observe any sign of spontaneous colitis. Since IL-10 KO in mice leads to enhanced pathological score upon DSS-mediated induction of colitis, and several colitis models are well established in birds, it will be worthwhile to test the consequence of experimentally inducing colitis in this context.

One of the novel features of this study is the observation that the microbiome is modified in the IL10KO HOM chicks, which may serve to mitigate potential spontaneous pathology; we now mention this in the discussion. We agree that it could be worthwhile in the future to look at additional challenge models. However, we would argue that the *Eimeria* challenge is a sufficiently adequate experimentally-induced model of colitis to demonstrate the increased inflammation that occurs in an IL10-deficient bird. This is further supported by evidence of enhanced inflammatory responses in the caeca of IL10KO HOM birds challenged with *Campylobacter* or *Salmonella* relative to WT controls. See in the revised manuscript (pages 12-13).

Minor points:(1) In Figure 2B, the authors should confirm whether the ROS-AV163 groups also have LPS treatment.

The legend for Figure 2B already states that neutralizing anti-IL10 antibody was added to LPS-stimulated BMDMs: “Nitric oxide production was assessed by measuring nitrite levels using Griess assay for LPS-stimulated BMDMs […] in the absence or presence of neutralizing anti-IL10 antibody ROS-AV163”. However, for added clarity we have now modified the x-axis label for Figure 2B (“+ROS-AV163” replaced by “+LPS +anti-IL10”) and we have also made minor changes to the figure legend. See in the revised manuscript (page 33).

(2) In Figure 3F, the authors should discuss why the duodenum of KO birds has enhanced infiltration compared to WT?

We are not sure what the reviewer is referring to here. Although not specifically mentioned in Figure 3F, there is no statistically significant difference in cellular infiltration in the duodenum of IL10KO WT and HOM birds raised in our specified pathogen-free (SPF) facility, nor in the duodenum of IL10KO WT and HOM birds raised in our conventional facility (Mann-Whitney U tests, p>0.1 in both cases); this can be seen in the sums of histopathological scores shown in Figures 3C (SPF facility) and 3E (conventional facility). Figure 3F shows that there is a statistically significant difference in cellular infiltration scores in the duodenum and proximal colon of both IL10KO WT and HOM birds based on the environment they are raised in (SPF vs conventional). We have made minor changes to the text to clarify this. See in the revised manuscript (page 7).

(3) The authors should discuss the observed differences in the C. jejuni colonization results among the two cohorts at week 1 and week 2 post-infection.

Numbers of C. jejuni in the caeca of IL10KO HOM birds were markedly lower than for WT controls at 1-week post-infection in cohort 1, and at both time intervals post-infection in cohort 2 (Figure 4A). This reached statistical significance at 1-week post-infection in cohort 1 and at 2-weeks post-infection in cohort 2. It is evident from Figure 4A that considerable inter-animal variance existed in each group, and in the IL10KO HOM birds in particular. This is typical of C. jejuni colonisation in chickens, where bacterial population structures have been reported to be variable and unpredictable (Coward et al., Appl Environ Microbiol 2008, PMID: 18424530). Similar variation between time intervals, birds and repeated experiments has been reported when evaluating vaccines against C. jejuni colonisation (e.g. Buckley et al., Vaccine 2010, PMID: 19853682; Nothaft et al., Front Microbiol 2021, PMID: 34867850). We performed two independent studies for this reason. Taken together, we consider that our data provide convincing evidence of elevated pro-inflammatory responses upon C. jejuni infection in IL10KO HOM birds relative to WT controls that associates with reduced bacterial burden. Our data is also consistent with a published observation that a commercial broiler line with low IL10 expression had correspondingly elevated expression of CXCLi-1, CXCLi-2 and IL-1b (Humphrey et al., mBio 2014, reference 33 in our original submission). We have added text to the discussion to capture the points above. See in the revised manuscript (page 13).

**Reviewer #2 (Recommendations for the authors):**
For the animal challenging experiments, both IL10KO HOM and IL10EnKO HOM chickens were used for *Eimeria* challenge, but not for *Salmonella* and *Campylobacter*. Could the authors justify why?

The *Eimeria* challenge produced a much higher and more reproducible level of inflammation than either of the bacterial challenge models. Within the parasite challenge cohorts, IL10KO HET and IL10EnKO HOM birds were only marginally different from WT controls (e.g. parasite replication: Figures 5A and B; lesion scores: Figures 5E and F; body weight gain: Figures 5G and H). Given the more limited response and the inter-individual variation in the bacterial challenge models, we felt that analysis of a sufficiently large cohort of the IL10KO HOM was appropriate, while additional cohorts of IL10KO HET and IL10EnKO HOM birds large enough to detect statistically significant differences could not be justified.

In the M&M, there was no mention of # of birds generated for IL10EnKO HOM, HET, etc.

Full details of bird numbers can be found in SI Appendix Table S1 “Number of IL10KO and IL10EnKO WT, HET and HOM chicks hatched in the NARF SPF chicken facility in the first (G1) and second (G2) generations”. Table S1 is already referred to in the Results section “Generation of IL10-deficient chickens”; we have now also clearly referred to it in the “Animals” and “Generation of surrogate host chickens and establishment of the IL10KO and IL10EnKO lines under SPF conditions” sections of the Materials and Methods. In all three sections we have also added some text to clarify that the table details G1 and G2 bird numbers. See in the revised manuscript (pages 5, 15, 17).

From the results of Campylobacter challenge, the results from the cohort 1 and cohort 2 were not consistent at both 1 and 2 weeks of post-infection. There is not much discussion on this inconsistency. What is the final conclusion: significant difference in week 1 or week 2, OR none of them, OR both of them. What would happen if an additional cohort were conducted for *Salmonella* and *Eimeria*?

As noted in response to Reviewer 1 (minor point 3), we have now added text to the discussion on the partial inconsistency between independent C. jejuni challenge studies. We do not feel that additional experiments to address this comment are required. Highly significant increases in the infiltration of lymphoplasmacytic cells and heterophils were detected in IL10KO HOM chickens relative to WT controls in the caeca, a key site of Campylobacter colonisation. This was consistently observed in two independent cohorts at both 1- and 2-weeks post-infection (SI Appendix Figures S7 and S8) and was reflected in similar patterns of expression of pro-inflammatory genes at these intervals in both cohorts (Figure 4B). As our laboratory has observed substantially less variation between repeated *Salmonella* challenges, a single study was performed, but with adequate power to detect statistical differences. The effects of E. tenella infection in IL10KO WT and HOM birds were replicated (compare Figure 4 with data from day 6 in Figure 5).